# mTORC1 couples cyst(e)ine availability with GPX4 protein synthesis and ferroptosis regulation

Yilei Zhang[1], Robert V. Swanda[2], Litong Nie [1], Xiaoguang Liu [1], Chao Wang[1], Hyemin Lee[1], Guang Lei [1], Chao Mao[1], Pranavi Koppula [1,3], Weijie Cheng[1], Jie Zhang[1], Zhenna Xiao[4], Li Zhuang[1], Bingliang Fang[5], Junjie Chen [1], Shu-Bing Qian [2,6] & Boyi Gan [1,3✉]

Glutathione peroxidase 4 (GPX4) utilizes glutathione (GSH) to detoxify lipid peroxidation and plays an essential role in inhibiting ferroptosis. As a selenoprotein, GPX4 protein synthesis is highly inefficient and energetically costly. How cells coordinate GPX4 synthesis with nutrient availability remains unclear. In this study, we perform integrated proteomic and functional analyses to reveal that SLC7A11-mediated cystine uptake promotes not only GSH synthesis, but also GPX4 protein synthesis. Mechanistically, we find that cyst(e)ine activates mechanistic/mammalian target of rapamycin complex 1 (mTORC1) and promotes GPX4 protein synthesis at least partly through the Rag-mTORC1-4EBP signaling axis. We show that pharmacologic inhibition of mTORC1 decreases GPX4 protein levels, sensitizes cancer cells to ferroptosis, and synergizes with ferroptosis inducers to suppress patient-derived xenograft tumor growth in vivo. Together, our results reveal a regulatory mechanism to coordinate GPX4 protein synthesis with cyst(e)ine availability and suggest using combinatorial therapy of mTORC1 inhibitors and ferroptosis inducers in cancer treatment.

[1] Department of Experimental Radiation Oncology, the University of Texas MD Anderson Cancer Center, Houston, TX, USA. [2] Graduate Field of Biomedical and Biological Sciences, Cornell University, Ithaca, NY, USA. [3] The University of Texas MD Anderson UTHealth Graduate School of Biomedical Sciences, Houston, TX, USA. [4] Department of Genetics, the University of Texas MD Anderson Cancer Center, Houston, TX, USA. [5] Department of Thoracic and Cardiovascular Surgery, the University of Texas MD Anderson Cancer Center, Houston, TX, USA. [6] Division of Nutritional Sciences, Cornell University, Ithaca, NY, USA. ✉email: bgan@mdanderson.org

Ferroptosis is an iron-dependent form of regulated cell death that is caused by excessive lipid peroxidation. It is morphologically, biochemically, and genetically distinct from other forms of regulated cell death[1,2]. Ferroptosis is inhibited by glutathione peroxidase 4 (GPX4), a glutathione peroxidase that utilizes reduced glutathione (GSH) as a cofactor to detoxify lipid peroxidation[3,4]. Glutathione is a tripeptide that is derived from cysteine, glutamate, and glycine, among which cysteine is the rate-limiting precursor. Most cancer cells obtain cysteine primarily through the amino-acid transporter solute carrier family 7 member 11 (SLC7A11; also known as xCT)-mediated uptake of extracellular cystine (which is an oxidized dimeric form of cysteine), followed by the reduction of intracellular cystine to cysteine[5–8]. Cysteine is subsequently utilized for glutathione biosynthesis as well as for protein synthesis and other metabolic processes[9,10]. Cystine starvation in culture medium or treatment with different classes of ferroptosis inducers (FINs) induces ferroptosis[2–4]. For example, classes 1 or 2 FINs induce ferroptosis by blocking SLC7A11-mediated cystine transport or inactivating GPX4, respectively[11]. Compared with cystine starvation or FIN treatment, GSH depletion generally results in much milder ferroptosis phenotype (or even does not induce obvious cell death in some cell lines)[12], indicating that there might exist additional mechanisms linking SLC7A11-mediated cystine transport to GPX4 function in ferroptosis regulation.

Selenocysteine, the 21st proteinogenic amino acid, is structurally similar to cysteine, except the sulfur in cysteine is replaced with selenium in selenocysteine. Because selenocysteine has lower pKa and reduction potential than cysteine, a few critical proteins with redox-related functions have selenocysteine instead of cysteine in their key residues involved in redox reactions[13]. The human genome encodes 25 selenocysteine-containing proteins (selenoproteins), including GPX4[14]. Of all the selenoproteins that have been individually deleted in mouse, only *Gpx4* KO mice exhibit an embryonic lethal phenotype similar to that of mice with selenocysteine tRNA gene deletion[15,16]. In addition, as long as partial GPX4 activity is maintained, cells deficient in all other selenoproteins can still survive and proliferate[17]. These studies suggest that GPX4 appears to be the most important selenoprotein at least in some cellular contexts. Selenocysteine is synthesized and incorporated into selenoproteins through the binding of selenocysteine tRNA at the opal codon UGA via a highly complex and energetically costly process[13]. Considering that selenoprotein synthesis is highly inefficient and energetically costly and that GPX4 is critical in preventing ferroptosis, GPX4 protein synthesis needs to be tightly controlled. However, the mechanisms by which GPX4 protein synthesis is regulated remain poorly understood.

Protein synthesis is a highly energy-consuming process and therefore needs to be tightly coordinated with nutrient and energy availability. One key signaling node that integrates a wide range of environmental cues to regulate protein synthesis is mechanistic target of rapamycin complex 1 (mTORC1, also known as mammalian TORC1)[18–21]. mTORC1 exists as a multiprotein complex consisting of mTOR, Raptor, and other proteins[18,19]. mTORC1 can be potently activated by amino acids, growth factors, or glucose (which provides ATP as the energy source), among other stimuli. Once activated, mTORC1 promotes protein synthesis through a variety of downstream effectors, prominent among which are p70S6 Kinase (S6K) and eukaryotic initiation factor 4E (eIF4E)-binding proteins (4EBPs). Upon phosphorylation by mTORC1, S6K phosphorylates ribosomal protein S6 as well as other substrates to promote mRNA translation. 4EBPs bind to eIF4E and inhibit eIF4E-mediated translation initiation. Phosphorylation of 4EBPs by mTORC1 releases 4EBPs from eIF4E, thereby allowing 5′-cap-dependent translation initiation[22].

Besides controlling protein synthesis, mTORC1 also regulates many other cellular processes involved in cell growth and metabolism, such as lipid metabolism and autophagy[18]. However, whether and how mTORC1 regulates ferroptosis remain largely unknown. In this study, we show that cyst(e)ine not only promotes GSH biosynthesis, but also promotes GPX4 protein synthesis through activating mTORC1 (in this manuscript, we use the term "cyst(e)ine" to refer to "cystine and cysteine"), and that mTORC1 inactivation sensitizes cancer cells to ferroptosis by decreasing GPX4 synthesis, therefore revealing a crosstalk between mTORC1 and ferroptosis.

## Results

**SLC7A11-mediated cystine uptake promotes GPX4 protein synthesis.** To characterize proteomic alterations associated with cystine starvation in cancer cells, we conducted comparative proteomic analyses in UMRC6 cells cultured in control (200 μM) or cystine-low (cystine starvation; 1 μM cystine) medium (Supplementary Fig. 1a). A principal component analysis based on all proteins quantified under both conditions showed a clear separation between control and cystine starvation conditions (Supplementary Fig. 1b). Our proteomic analysis identified 16 and 12 proteins with significantly increased or decreased expression upon cystine starvation, respectively (Fig. 1a and Supplementary Table 1–2). Most of the top upregulated proteins, such as asparagine synthase and heme oxygenase 1, are involved in integrated stress responses and therefore are likely transcriptionally induced by cystine starvation by activating transcription factor 4 (ATF4)[23]. How cystine starvation suppresses various identified proteins remains unclear. Notably, GPX4 was among the top downregulated proteins upon cystine starvation (Fig. 1a). Given the important roles of both cyst(e)ine and GPX4 in inhibiting ferroptosis, we focused on this regulation in our following studies.

Western blotting confirmed that cystine starvation significantly decreased GPX4 protein levels in UMRC6 cells; of note, cystine starvation did not affect the levels of other ferroptosis regulators, such as acyl coenzyme A synthetase long-chain family member 4 (ACSL4) and ferroptosis suppressor protein 1 (FSP1) (Fig. 1b). Consistent with our and others' previous reports[2,24], cystine starvation, similar to erastin treatment, potently induced SLC7A11 expression (Fig. 1b), possibly through ATF4 and/or Nrf2-mediated transcriptional adaptive response under cystine-limiting conditions. In line with this, other studies have shown that SLC7A11 expression can be induced under other stress conditions[6,7,25]. Further analyses confirmed this observation in a variety of cell lines and revealed that cystine starvation did not cause a corresponding decrease of *GPX4* mRNA levels (if any, cystine deprivation led to moderately increased *GPX4* mRNA levels) (Fig. 1c and Supplementary Fig. 1c–g).

It is important to note that, at the time points when we observed GPX4 level reduction under cystine starvation, there was no obvious ferroptosis induction (or any cell death) in these cell lines (most of which, such as UMRC6 and NCI-H226 cells[24], exhibit high SLC7A11 expression and are ferroptosis resistant). (In contrast, in some other cell lines that are exquisitely sensitive to ferroptosis, cells rapidly died upon cystine starvation before we could observe potential GPX4 level change.) This prompted us to further examine the role of SLC7A11, a major cystine transporter, in regulating GPX4 levels. We found that treatment with erastin, a class 1 FIN, or genetic ablation of *SLC7A11* in UMRC6 cells dramatically decreased cystine uptake and GPX4 protein levels without reducing *GPX4* mRNA levels (Fig. 1d–g and Supplementary Fig. 1h–l). Conversely, SLC7A11 overexpression in cell lines with low endogenous SLC7A11 expression significantly

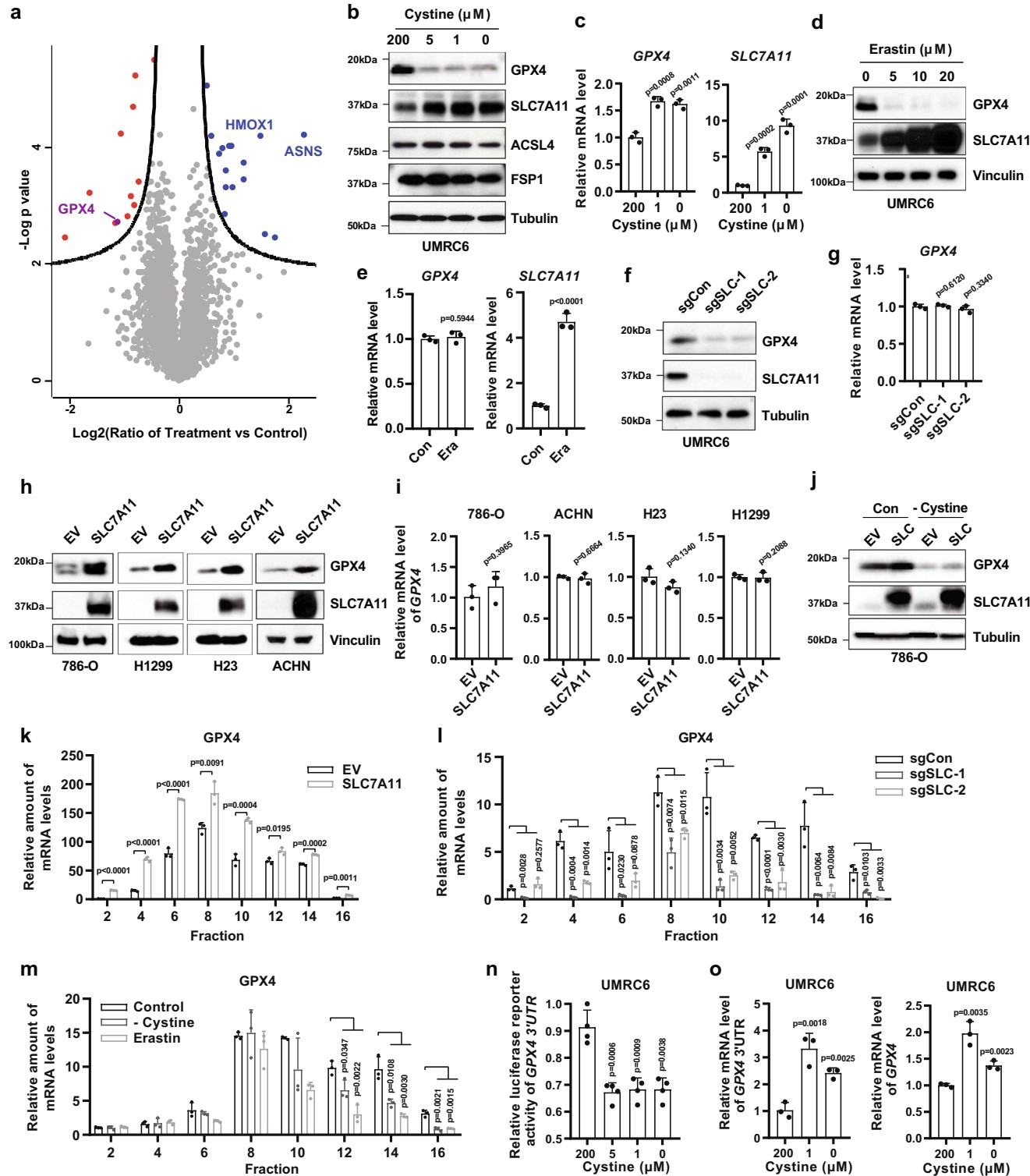

increased GPX4 protein levels without affecting *GPX4* mRNA levels (Fig. 1h, i). Furthermore, cystine deprivation or erastin treatment abolished SLC7A11 overexpression–induced GPX4 expression (Fig. 1j and Supplementary Fig. 1m–o).

A decrease in GPX4 protein levels without a corresponding decrease in its mRNA levels could indicate increased protein degradation or decreased protein synthesis (or both). Treatment with the proteasome inhibitor MG132 or the lysosome inhibitor chloroquine or both did not restore GPX4 protein levels under cystine deprivation or erastin treatment or in SLC7A11-

KO cells (as a control, MG132 increased p53 or HIF2α protein levels, and chloroquine increased LC3 form II levels) (Supplementary Fig. 2a–c). A previous study suggested involvement of lysosome-associated membrane protein 2 (LAMP2) and chaperone-mediated autophagy in the degradation of GPX4 upon erastin treatment[26]. However, *LAMP2* deletion in UMRC6 cells did not affect the reduction of GPX4 protein levels under erastin treatment or cystine starvation (Supplementary Fig. 2d), suggesting that, at least in the cell line we have examined, cystine starvation regulates GPX4 protein levels independent of LAMP2.

**Fig. 1 SLC7A11-mediated cystine uptake promotes GPX4 protein synthesis. a** UMRC6 cells were cultured in control (200 μM) or treatment (1 μM cystine) media for 30 h followed by MS analysis for regulated proteins. Volcano plot shows the differentially expressed proteins in treatment vs control cells. Each filled circle represents a protein; significantly upregulated and downregulated proteins are highlighted with blue and red, respectively. The curve is derived at FDR = 0.05 and S0 = 0.1 as described in the Methods. **b** UMRC6 cells were cultured in media with indicated concentrations of cystine for 24 h. Protein levels were evaluated by Western blotting. **c** GPX4 and SLC7A11 mRNA levels were measured by RT-PCR in UMRC6 cells cultured in media with indicated concentrations of cystine for 24 h. n = 3. **d** UMRC6 cells were cultured in media with indicated concentrations of erastin for 24 h. GPX4 and SLC7A11 protein levels were evaluated by Western blotting. **e** GPX4 and SLC7A11 mRNA levels were measured by RT-PCR in UMRC6 cells treated with 5 μM erastin for 24 h. n = 3. **f** GPX4 and SLC7A11 protein levels were evaluated by Western blotting in control (sgCon) and SLC7A11-knockout (sgSLC-1 and sgSLC-2) UMRC6 cells. **g** GPX4 mRNA level was determined by RT-PCR in SLC7A11-knockout UMRC6 cells. n = 3. **h** GPX4 and SLC7A11 protein levels were evaluated by Western blotting in indicated SLC7A11-overexpressing cell lines. **i** GPX4 mRNA level was determined by RT-PCR in SLC7A11-overexpressing cell lines. n = 3. **j** Empty vector (EV) and SLC7A11-overexpressing (SLC) 786-O cells were cultured in complete (Con) or cystine-free (-cystine) media for 20 h followed by Western blotting to monitor GPX4 protein levels. **k** 786-O-EV and -SLC cell lines were subjected to polyribosome fractionation followed by RT-PCR to analyze GPX4 mRNA distribution profiles during protein translation. n = 3. **l** UMRC6-sgCon, -sgSLC-1, and -sgSLC-2 cells were subjected to polyribosome fractionation followed by RT-PCR to analyze GPX4 mRNA distribution profiles during protein translation. n = 3. **m** UMRC6 cells were cultured in control media, cystine-free media, or treated with erastin (10 μM) for 24 h followed by polyribosome fractionation and RT-PCR to analyze GPX4 mRNA distribution profiles during protein translation. n = 3. **n** Luciferase reporter activity for 3′-UTR of GPX4 gene was measured in UMRC6 cells treated with indicated concentrations of cystine for 24 h. n = 4. **o** RT-PCR analysis of 3′-UTR or exon region of GPX4 gene in UMRC6 cells treated with indicated concentrations of cystine for 24 h. n = 3. For all panels, error bars are mean ± SD. n indicates biologically independent repeats. P value was determined by two-tailed unpaired Student's t test. Source data are provided as a Source Data file.

In addition, cycloheximide chase analysis showed that cystine starvation did not decrease GPX4 protein half-life (Supplementary Fig. 2e). Therefore, it is less likely that the change in GPX4 protein levels in response to cystine starvation results from altered GPX4 protein degradation.

Next, we considered the possibility that cysteine regulates GPX4 protein synthesis. To test this, we subjected SLC7A11-overexpressing cells to sucrose gradient sedimentation, and then measured GPX4 transcripts (and Actin as a control) in different polysome fractions by real-time PCR. The analysis revealed that SLC7A11 overexpression promoted, whereas cystine deprivation or erastin treatment significantly decreased GPX4 (but not Actin) transcript enrichment in polysomes (Fig. 1k–m and Supplementary Fig. 2f–h). Furthermore, using a luciferase reporter assay in which we fused the luciferase reporter gene with GPX4 3′-UTR region, we showed that cystine deprivation significantly decreased luciferase activity; as a control, cystine deprivation even moderately increased GPX4 3′-UTR mRNA levels (Fig. 1n, o). Of note, puromycin incorporation and polysome fractionation analysis revealed that cystine starvation did not significantly decrease global protein synthesis or polysome profiles (Supplementary Fig. 2i–j), suggesting that cystine regulation of GPX4 protein synthesis is relatively specific. Taken together, our data strongly suggest that SLC7A11-mediated cystine uptake promotes GPX4 protein synthesis without affecting GPX4 mRNA levels or its protein degradation.

**SLC7A11 modulates ferroptosis sensitivity to class 2 FINs partly by regulating GPX4 levels.** Our data on SLC7A11 regulation of GPX4 protein synthesis also shed lights on the role of SLC7A11 in modulating ferroptosis sensitivity induced by GPX4 inhibitors. Specifically, analyses of the Cancer Therapeutics Response Portal (CTRP)[27–29] revealed that FSP1 (also known as AIFM2) exhibited the most striking positive correlation with resistance to multiple class 2 FINs that inactivate GPX4, including RSL3, ML162, and ML210 (Supplementary Fig. 3a), which is consistent with recent reports that FSP1 acts in parallel to GPX4 to inhibit ferroptosis[30,31]. The same analyses also showed that SLC7A11 and SLC3A1 (a SLC7A11 chaperone) were among the top genes whose expression positively correlates with the resistance to these GPX4 inhibitors (Supplementary Fig. 3a). We confirmed that SLC7A11 deficiency promoted, whereas SLC7A11 overexpression inhibited class 2 FINs-induced lipid peroxidation and ferroptosis (Fig. 2a–f and Supplementary Fig. 3b–e). Similar

to SLC7A11 deficiency, we found that erastin treatment or cystine starvation also sensitized cancer cells to ferroptosis induced by class 2 FINs (Fig. 2g and Supplementary Fig. 3f–h).

Based on the model that SLC7A11 operates upstream of GPX4, one would expect that SLC7A11 inactivation, similar to GSH depletion, should not affect ferroptosis sensitivity to class 2 FINs that inactivate GPX4 (in contrast, FSP1 functions in parallel to GPX4, consistent with the observation that FSP1 levels modulate ferroptosis sensitivity to GPX4 inhibitors[30,31]). In light of our data revealing SLC7A11 regulation of GPX4 protein levels, we reasoned that the differential sensitivities to class 2 FINs in SLC7A11-KO or -overexpressing cells could potentially be at least partly explained by differential GPX4 protein levels in these cells. To test this hypothesis, we decreased GPX4 levels in SLC7A11-overexpressing cells slightly higher than those in control cells by shRNA-mediated knockdown (Fig. 2h), and showed that GPX4-knockdown in SLC7A11-overexpressing cells at least partially re-sensitized these cells to RSL3 (Fig. 2i). Conversely, GPX4 restoration in SLC7A11-KO cells rendered these cells more resistant to RSL3- or ML162-induced ferroptosis (Fig. 2j–l and Supplementary Fig. 3i–j). Our data, therefore, suggest that SLC7A11 modulates ferroptosis sensitivity to class 2 FINs at least partly through regulating GPX4 protein levels.

**GSH depletion does not regulate GPX4 protein levels or ferroptosis sensitivity to class 2 FINs.** The aforementioned data prompted us to study how SLC7A11-mediated cystine transport promotes GPX4 protein synthesis. Once transported into cells, intracellular cystine is reduced to cysteine, which is then utilized in GSH biosynthesis (as well as other cellular processes such as protein synthesis). GSH biosynthesis is initiated by the condensation of cysteine and glutamate to form γ-glutamylcysteine (γ-Glu-Cys), which is mediated by glutamate–cysteine ligase (GCL), the rate-limiting enzyme in GSH biosynthesis that comprises a catalytic subunit (GCLC) and a modifying subunit (GCLM) (Supplementary Fig. 4a). Cystine starvation or erastin treatment is known to potently deplete intracellular GSH levels[3,32]. However, we found that in contrast to cystine starvation or erastin treatment, treatment with GCLC inhibitor l-buthionine sulfoximine (BSO) or GCLC knockdown, despite significantly decreasing intracellular GSH levels, did not decrease GPX4 protein levels or affect ferroptosis sensitivity to class 2 FINs (Supplementary Fig. 4b–j). Furthermore, supplementation with GSH ethyl ester (GSHEE, a membrane/lipid permeable derivative of

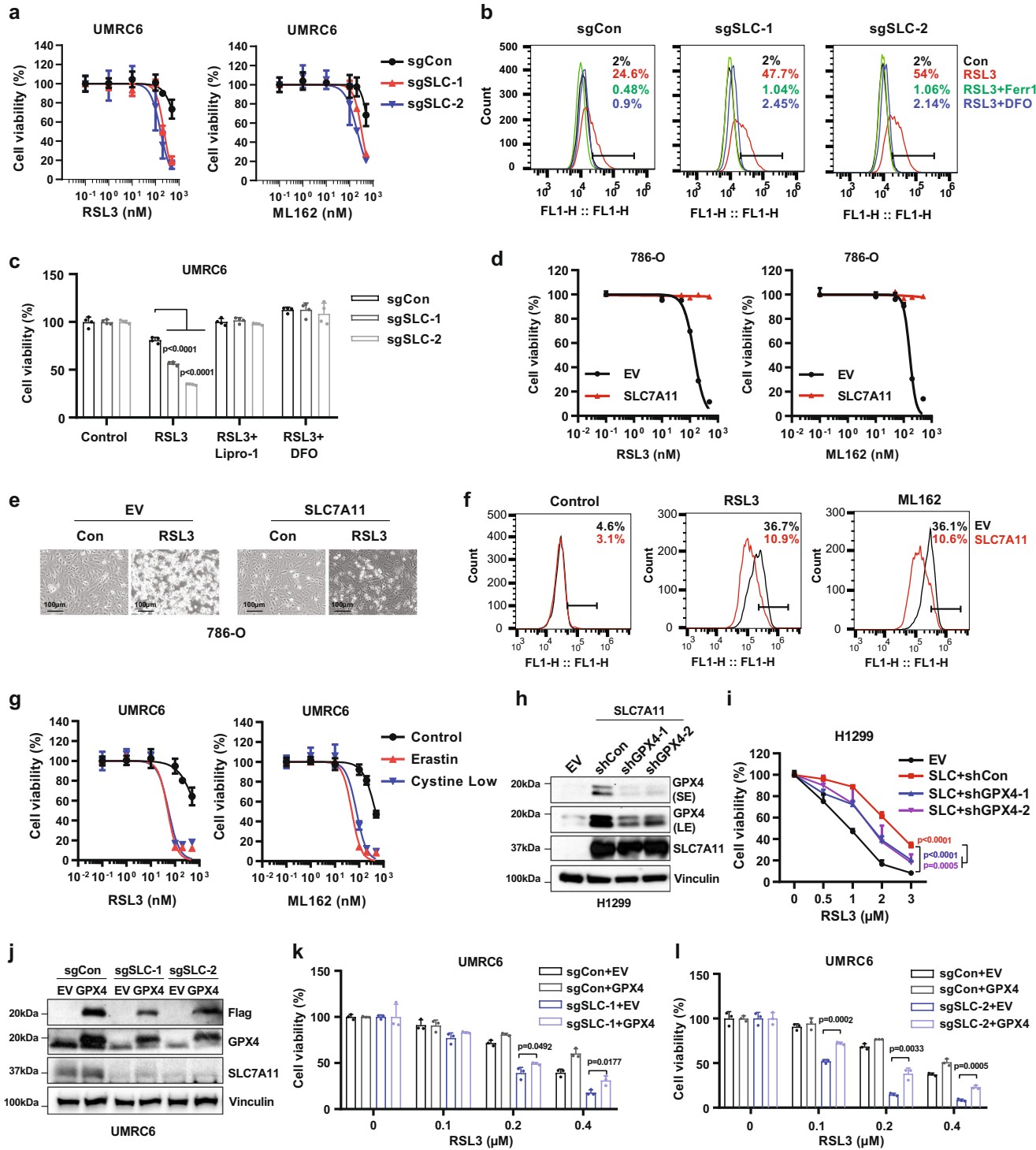

GSH) restored intracellular GSH levels but failed to restore GPX4 levels (Supplementary Fig. 4k–m) or affect ferroptosis sensitivity in *SLC7A11*-KO UMRC6 or SLC7A11-low 786-O cells (Supplementary Fig. 4n–o). Together, our data show that, unlike cystine starvation or erastin treatment, GSH depletion does not regulate GPX4 levels or ferroptosis sensitivity to class 2 FINs, suggesting that cyst(e)ine promotes GPX4 protein synthesis likely through GSH-independent mechanisms.

**Cyst(e)ine promotes GPX4 protein synthesis partly through Rag-mTORC1-4EBP signaling.** We reasoned that cyst(e)ine regulation of GPX4 protein synthesis might involve amino-acid-

sensing mechanisms, and tested whether mTORC1 is involved in cyst(e)ine regulation of GPX4 protein synthesis, considering that mTORC1 signaling represents a major amino-acid-sensing pathway to control protein synthesis[18,19]. We found that cystine starvation suppressed mTORC1 activation (Fig. 3a), whereas adding back cystine in cystine-free medium re-activated mTORC1 signaling (Supplementary Fig. 5a). In contrast, blocking GSH biosynthesis by BSO treatment, unlike cystine starvation, did not suppress mTORC1 activation (Supplementary Fig. 5b). Previous studies showed that mTORC1 activation by amino acids, particularly leucine and arginine, involves mTORC1 localization on lysosomes[18,19]. Consistent with this, we found that cystine

**Fig. 2 SLC7A11 modulates ferroptosis sensitivity to class 2 FINs partly through regulating GPX4 levels. a** UMRC6 cells were treated with RSL3 or ML162 at indicated concentrations for 10 h followed by cell viability analysis. **b** UMRC6 cells were treated with 600 nM RSL3 combined with or without 5 µM Ferrostatin-1 (Ferr-1) or 100 µM deferoxamine (DFO) for 6 h. Then lipid peroxidation was assessed using BODIPY™ 581/591 C11 staining followed by FACS analysis. **c** Cell viability was determined for control (sgCon) and *SLC7A11*-knockout (sgSLC-1 and sgSLC-2) UMRC6 cells treated with 500 nM RSL3 combined with or without 5 µM Ferrostatin-1 (Ferr-1) or 100 µM deferoxamine (DFO) for 10 h. *n* = 4. **d** Empty vector (EV) and SLC7A11-overexpressing (SLC7A11) 786-O cells were treated with RSL3 or ML162 at indicated concentrations for 9 h followed by cell viability analysis. **e** Representative photos of 786-O-EV and -SLC7A11 cells treated with 400 nM RSL3 for 9 h. **f** 786-O-EV and -SLC7A11 cells treated with 400 nM RSL3 for 6 h. Then lipid peroxidation was assessed using BODIPY™ 581/591 C11 staining followed by FACS analysis. **g** UMRC6 cells were treated with RSL3 or ML162 at indicated concentrations in the presence of 5 µM erastin or 1 µM cystine (Cystine low) for 10 h followed by cell viability analysis. **h** Western blotting analysis of GPX4 expression in H1299-EV, -SLC7A11, and H1299-SLC7A11 *GPX4*-knockdown cell lines. *SE* short exposure, *LE* long exposure. **i** Cell viability measured in cell lines described in **h** treated with indicated concentrations of RSL3 for 6 h. **j** Western blotting analysis of GPX4 expression in control (sgCon) and *SLC7A11*-KO (sgSLC-1 and sgSLC-2) UMRC6 cell lines overexpressed with empty vector (EV) or GPX4. **k**, **l** Cell viability measured in cell lines described in **j** treated with indicated concentrations of RSL3 for 10 h. *n* = 3. For all panels, error bars are mean ± SD. *n* indicates biologically independent repeats. *P* value was determined by two-tailed unpaired Student's *t* test. For **i**, *p* value was determined by two-way ANOVA test. Source data are provided as a Source Data file.

starvation significantly decreased mTOR localization on lysosomes (as demonstrated by the colocalization of mTOR with the lysosome marker protein LAMP2) and re-addition of cystine in cystine-free medium restored mTOR localization on lysosomes (Supplementary Fig. 5c–d).

We then examined whether mTORC1 inactivation, like cystine deprivation, decreases GPX4 protein levels. We tested two commonly used mTORC1 inhibitors: rapamycin, an allosteric mTORC1 inhibitor, and Torin1, a potent and selective ATP-competitive mTOR inhibitor[33]. Previous studies showed that Torin1 fully suppresses mTORC1, whereas rapamycin only partially inhibits it[33]. We found that, although both inhibitors were effective in suppressing S6K or S6 phosphorylation, Torin1 had a much more potent inhibitory effect than rapamycin on 4EBP1 phosphorylation; notably, treatment with Torin1, but not rapamycin, decreased GPX4 protein levels (Fig. 3b). Further analysis showed that Torin1 did not affect *GPX4* mRNA levels (Supplementary Fig. 5e). We confirmed that Torin1 decreased GPX4 protein levels in additional cell lines (Supplementary Fig. 5f–g). Similarly, GPX4 protein levels were decreased upon treatment with AZD8055 (Fig. 3c), another ATP-competitive mTOR inhibitor that potently inhibits both S6 and 4EBP1 phosphorylation[34]. We further showed that Torin1 or AZD8055 treatment significantly decreased *GPX4* transcript enrichment in polysomes (Fig. 3d). Consistently, Torin1 or AZD8055 treatment also significantly inhibited polysome formation during protein synthesis (Supplementary Fig. 5h). It should be noted that Torin1 and AZD8055 are mTOR kinase inhibitors that target both mTORC1 and mTORC2. However, deletion of *RICTOR*, a critical component of mTORC2, significantly reduced AKT phosphorylation as expected but did not affect GPX4 levels (Supplementary Fig. 5i), therefore ruling out the involvement of mTORC2 in regulating GPX4.

Tuberous sclerosis complex (TSC) genes TSC1 and TSC2 form a complex and are negative regulators of mTORC1; deficiency of *TSC1* or *TSC2* largely abolishes mTORC1 regulation by growth factors or energy levels, but not by amino acids[35], whereas amino-acid-mediated mTORC1 activation is largely mediated by Rags[18,19]. We found that *TSC1* deletion had no obvious effect on mTORC1 inactivation or the reduction of GPX4 protein levels under cystine starvation (Supplementary Fig. 5j–k); in contrast, *RagA/B* deletion largely abolished cystine stimulation-induced mTORC1 activation and GPX4 expression (Fig. 3e, f), suggesting that cystine regulates mTORC1 and GPX4 through Rag but independent of TSC.

Our data using different mTORC1 inhibitors suggested that mTORC1 inhibition likely decreases GPX4 protein levels through the 4EBP axis downstream of mTORC1. In support of this hypothesis, we showed that, like Torin1 treatment, doxycycline-

induced expression of 4EBP1 4 A (a non-phosphorylatable mutant of 4EBP1 in which all four mTORC1-sensitive phosphorylation sites were mutated to Ala, resulting in constitutive 4EBP1 binding to eIF4E[36]) decreased GPX4 protein levels (Fig. 3g, h). In addition, *4EBP1*/2-double-knockout (DKO) largely abolished the reduction of GPX4 protein levels under Torin1 treatment or cystine starvation (Fig. 3i–k). Tissue microarray analysis revealed that phosphor 4EBP1 staining correlated with GPX4 level, but not ACSL4 level, in tumor samples from cancer patients (Fig. 3l, m and Supplementary Fig. 5l).

eIF2α also has an important role in coordinating amino-acid availability with translational control through an amino-acid-induced dephosphorylation of eIF2α[37]. Consistent with this, we observed that cystine starvation significantly increased eIF2a phosphorylation in UMRC6 cells (Supplementary Fig. 5m). We further tested whether inducing eIF2α phosphorylation by treatment with salubrina (an inhibitor of eIF2α phosphatase) in the presence of cystine would mimic cystine starvation to decrease GPX4 levels. As shown in Supplementary Fig. 5n, whereas both cystine starvation and salubrina treatment increased eIF2α phosphorylation, cystine starvation, but not salubrina treatment, significantly decreased GPX4 levels. These data therefore suggest that cyst(e)ine regulates GPX4 protein levels likely through eIF2α-independent mechanisms. Together, our data suggest that cyst(e)ine stimulation activates mTORC1 through Rag-dependent mechanisms; once activated, mTORC1 promotes GPX4 protein synthesis likely through the downstream 4EBP axis (Supplementary Fig. 5o).

**mTORC1 inhibition sensitizes cancer cells or tumors to ferroptosis.** We next sought to determine whether mTORC1 inhibition, by decreasing GPX4 protein levels, can sensitize cancer cells to ferroptosis induced by GPX4 inhibitors. We showed that RSL3 or ML162 treatment induced lipid peroxidation and ferroptosis as expected; notably, although Torin1 treatment alone did not affect cell viability (at the time points when GPX4 inhibitors induced ferroptosis), it drastically sensitized UMRC6 cells to RSL3- or ML162-induced lipid peroxidation and ferroptosis (Fig. 4a, b and Supplementary Fig. 6a–b). We made similar observations in additional cell lines (Supplementary Fig. 6c–d). Furthermore, we showed that AZD8055, but not rapamycin, sensitized cancer cells to ferroptosis induced by class 2 FINs (Supplementary Fig. 6e–f), consistent with the differential effects of these mTORC1 inhibitors on 4EBP phosphorylation and GPX4 levels. Finally, we showed that combining Torin1 and BSO did not further induce lipid peroxidation or ferroptosis (Supplementary Fig. 6g–h). Our data, therefore, suggest that pharmacologic inhibition of mTORC1 sensitizes cancer cells to GPX4 inhibition-induced ferroptosis. Of note, to minimize the effect of

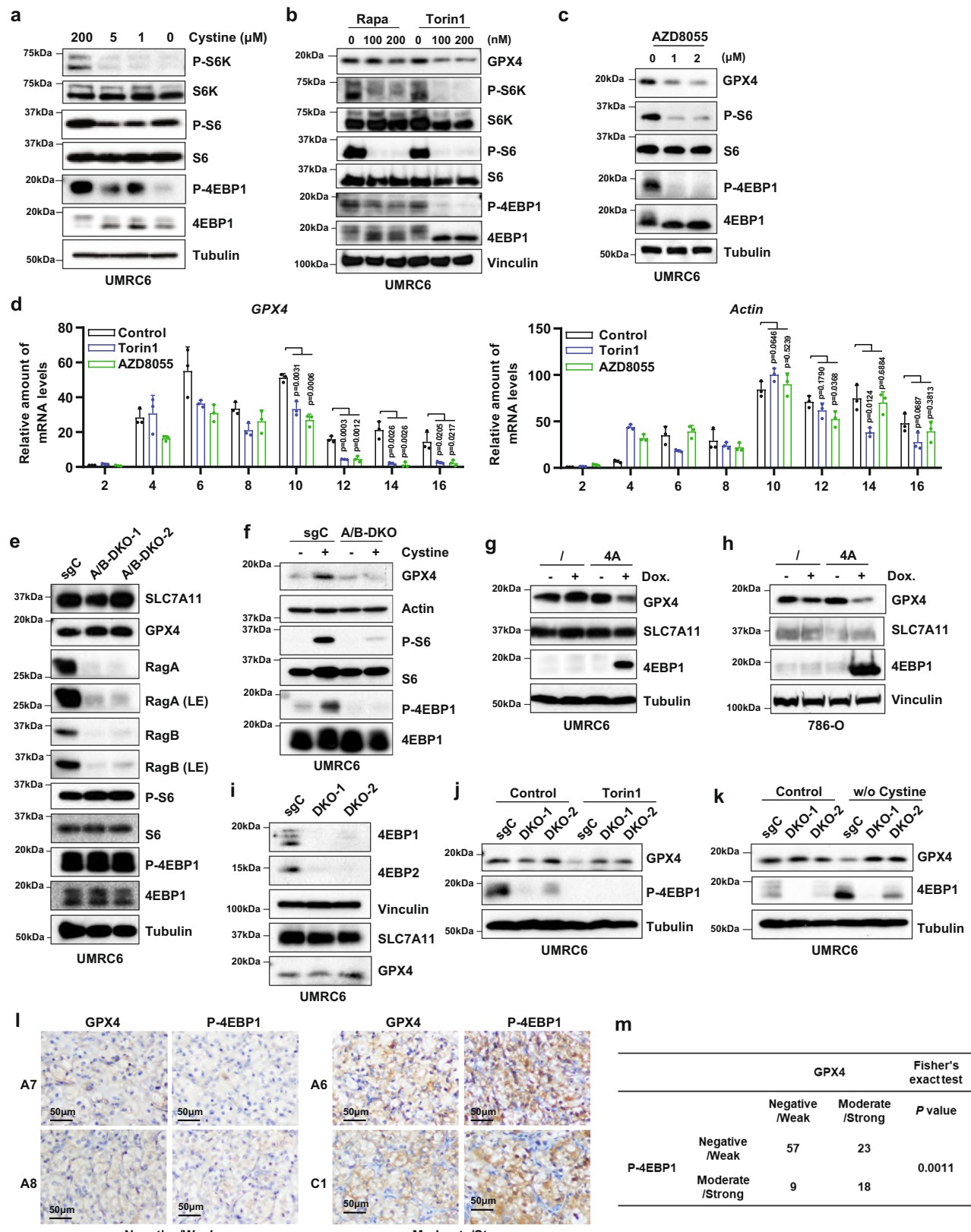

mTORC1 inhibitor treatment alone on cell proliferation and viability, we have focused on the cell viability/death measurement at earlier time points (such as 8 h) throughout our analyses. Long-term mTORC1 inhibition (such as 24 h) indeed exerted cytostatic and/or cytotoxic effects (Supplementary Fig. 6i), which is consistent with previous reports[33,34]; we further showed that the cell death caused by mTORC1 inhibition upon long-term treatment was not ferroptosis, as this cell death could not be rescued by ferroptosis inhibitor ferrostatin-1 or iron chelator DFO treatment (Supplementary Fig. 6i).

We also studied whether 4EBPs have a role in regulating ferroptosis sensitivity. *4EBP* DKO did not affect GPX4 protein

**Fig. 3 Cyst(e)ine promotes GPX4 protein synthesis partly through Rag-mTORC1-4EBP signaling. a** UMRC6 cells were cultured in media with indicated concentrations of cystine for 24 h followed by Western blotting to analyze protein levels. **b** UMRC6 cells were treated with indicated concentrations of Rapamycin (Rapa) or Torin1 for 24 h. GPX4 and mTOR signaling protein levels were assessed by Western blotting. **c** UMRC6 cells were treated with indicated concentrations of AZD8055 for 24 h. GPX4 and mTOR signaling protein levels were assessed by Western blotting. **d** UMRC6 cells were treated with 1 μM Torin1 or AZD8055 for 24 h followed by polyribosome fractionation and RT-PCR to analyze *GPX4* and *ACTB* (Actin) mRNA distribution profiles during protein translation. $n = 3$ biologically independent repeats. Error bars are mean ± SD. P value was determined by two-tailed unpaired Student's *t* test. **e** Western blotting analysis of indicated protein levels in UMRC6 control (sgC) and *RagA/RagB* double-knockout (A/B-DKO) cell lines. *LE* long exposure. **f** sgC and A/B-DKO cells were cultured in cystine-free media for 24 h followed by 200 μM cystine stimulation for 8 h and Western blotting analysis. **g, h** Wild-type (shown as "/") and 4EBP1-4A–overexpressing (shown as "4 A") UMRC6 or 786-O cells were treated with or without 100 ng/ml Doxycycline (Dox.) for 30 h followed by Western blotting analysis. **i** Western blotting analysis of protein expression in control (sgCon) and *4EBP1/2*-double-knockout (DKO-1 and DKO-2) UMRC6 cells. **j** Western blotting analysis of protein expression in indicated cell lines treated with or without 1 μM Torin1 for 24 h. **k** Western blotting analysis of protein expression in indicated cell lines cultured in complete media or cystine-free media for 24 h. **l** Representative images from immunohistochemical staining of kidney tumor tissue microarray for GPX4 and P-4EBP1. **m** Summary and statistical analysis of immunohistochemical staining results for GPX4 and P-4EBP1 in tumor tissue microarray.

levels under basal conditions but largely abolished the reduction of GPX4 protein levels under Torin1 treatment or cystine starvation (see Fig. 3i–k). Consistent with this, we showed that (i) *4EBP* DKO did not affect ferroptosis sensitivity to GPX4 inhibitors (Supplementary Fig. 6j), but largely restored the sensitizing effect of Torin1 to GPX4 inhibition-induced ferroptosis (Fig. 4c, d); and (ii) *4EBP* DKO rendered cells more resistant to cystine starvation- or erastin-induced ferroptosis (Fig. 4e, f).

We further tested the therapeutic potential of combining mTORC1 inhibitors and FINs in tumor treatment using patient-derived xenografts (PDXs). Because none of current GPX4 inhibitors is suitable for in vivo treatment, we have used imidazole ketone erastin (IKE) as the FIN in our animal studies; IKE is an erastin analog that potently blocks SLC7A11 transporter activity and it was shown recently that IKE is suitable for animal treatment and exhibits potent antitumor effects in xenograft models[38]. We showed that combined treatment with AZD8055 and IKE suppressed PDX tumor growth much more potently than did either treatment alone (Fig. 4g). AZD8055 and/or IKE treatment did not cause any significant weight loss in our animal studies, suggesting that these treatments were well-tolerated in vivo (Supplementary Fig. 6k). Further analyses revealed that treatment with IKE or AZD8055 alone only had very moderate effect on reducing phosphor 4EBP1 or GPX4 level in tumors, but their combination resulted in a potent suppression of phosphor 4EBP1 and GPX4 levels (Fig. 4h–i). The combination treatment also synergistically increased the staining of 4-hydroxy-2-noneal (4-HNE), a lipid peroxidation marker[39,40], in PDX tumor samples (Fig. 4h–i). Therefore, AZD8055 can synergize with IKE to suppress GPX4 levels in PDX tumors, thereby sensitizing tumors for IKE-induced ferroptosis in vivo. Together, our results show that mTORC1 inhibition sensitizes cancer cells or tumors to ferroptosis, and suggest combining mTORC1 inhibitors with FINs in cancer treatment.

## Discussion
In this study, we revealed that cyst(e)ine promotes the synthesis of not only GSH but also GPX4 protein. Mechanistically, we showed that cyst(e)ine promotes GPX4 protein synthesis independently of GSH but at least partly through the Rag-mTORC1-4EBP-signaling axis (Supplementary Fig. 7). Because selenoprotein synthesis is highly inefficient and energetically costly[13] and arguably, GPX4 is the most critical selenoprotein[17], GPX4 protein synthesis must be tightly controlled. We speculate that cells might have evolved a cyst(e)ine-sensing mechanism to coordinate the synthesis of both GPX4 protein and GSH, therefore enabling cells to precisely control GPX4 production based on the availability of its major cofactor. Of note, cysteine is one of the least-utilized proteinogenic amino acids in mammalian proteomes, likely

owing to its high biosynthetic energy cost[41]. Consistent with this, we showed that cystine starvation did not obviously inhibit global protein synthesis (Supplementary Fig. 2i), highlighting that cyst(e)ine regulation of GPX4 protein synthesis is relatively specific.

It is proposed that cyst(e)ine provides the rate-limiting precursor for the biosynthesis of GSH, which is subsequently used as the cofactor for GPX4-mediated lipid peroxidation detoxification and ferroptosis suppression[1]. However, in this study, we noted two major differences between cystine starvation and GSH depletion: (i) cystine starvation generally is much more potent than GSH depletion in inducing ferroptosis; (ii) cystine starvation, but not GSH depletion, sensitizes cells to class 2 FIN-induced ferroptosis. We propose that these differential effects can be explained by additional GSH-independent mechanisms downstream of cyst(e)ine in ferroptosis regulation, one of which is cyst(e)ine regulation of GPX4 protein synthesis as revealed in our current study. We want to emphasize that there exist additional ferroptosis defense mechanisms that operate in parallel to or independent of GPX4. For example, recent studies showed that FSP1 operates in parallel to GPX4 to inhibit ferroptosis by supplying coenzyme $Q_{10}$ (CoQ)[30,31] and that coenzyme A (CoA) is also capable of inhibiting ferroptosis independent of GPX4[42]. Given that cysteine is a precursor for CoA biosynthesis and that CoA is also utilized in CoQ biosynthesis[43], it is possible that these recently identified anti-ferroptisis pathways represent additional mechanisms linking cyst(e)ine to ferroptosis regulation independent of GPX4, which likely explains the lack of strong ferroptosis induction in cells treated with mTORC1 inhibitor and BSO (to suppress both GPX4 and GSH synthesis) (Supplementary Fig. 6h).

A recent study showed rapamycin treatment can decrease GPX4 protein levels[44], whereas in our study rapamycin treatment did not obviously affect GPX4 protein levels. We noticed that the rapamycin concentration used in this study (25 μM) was much higher than that used in our study (as well as in most other studies; typically in nM ranges). It is possible that rapamycin at such high concentrations can potently inhibit 4EBP1 phosphorylation and therefore suppress GPX4 protein synthesis. Another previous study indicated that mTORC1 regulates GPX4 protein translation[45]. However, this study showed that rapamycin treatment even increased GPX4 levels in the context of imatinib treatment. It remains unclear how mTORC1, a positive translation regulator, would suppress GPX4 protein synthesis in this context. Further studies are needed to clarify these questions.

Our recent study showed that AMP-activated protein kinase (AMPK) suppresses ferroptosis at least partly by inhibiting polyunsaturated fatty-acid (PUFA) biosynthesis[46], which is known to be required for lipid peroxidation and ferroptosis[47,48].

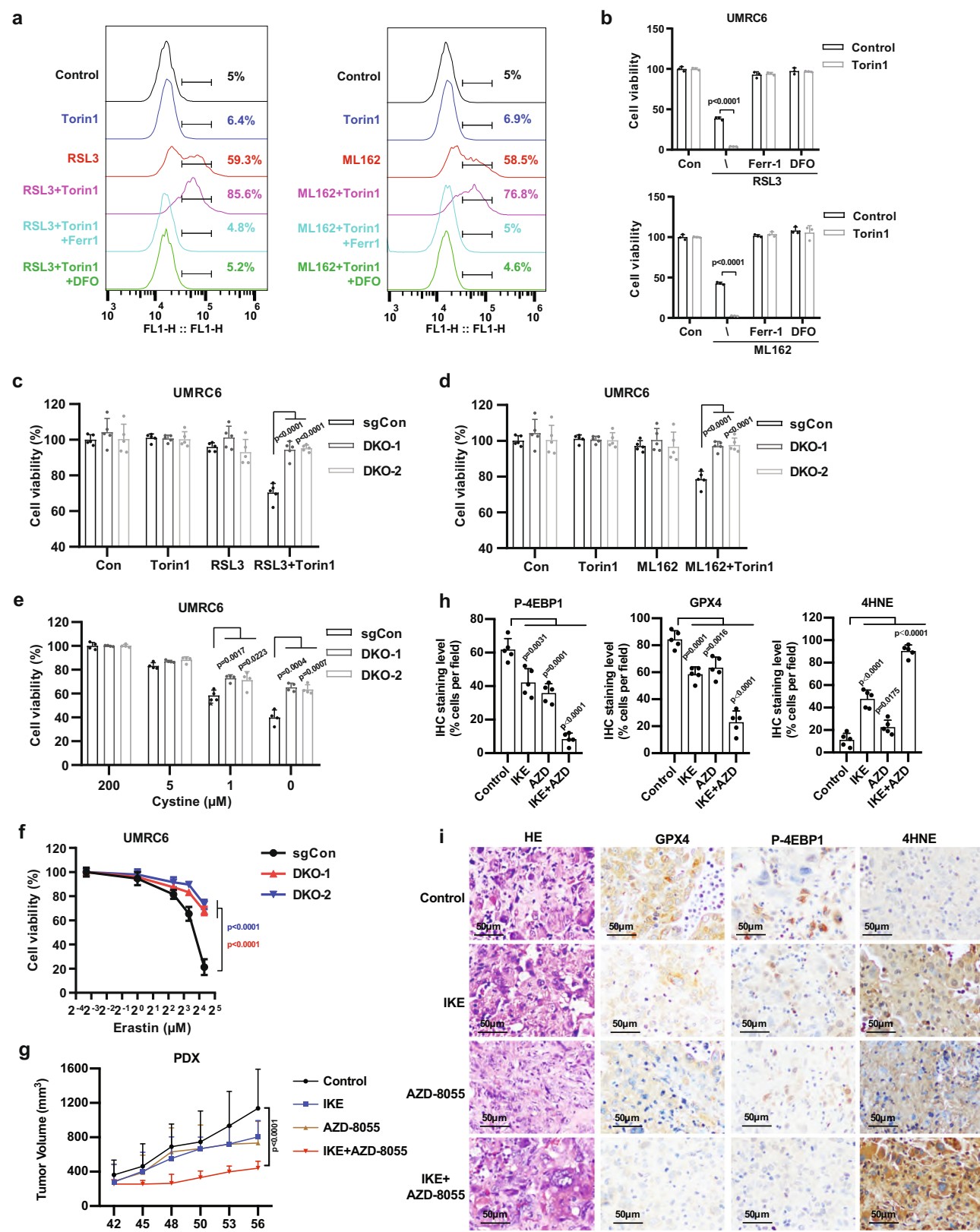

As AMPK inhibits mTORC1[49], it might be expected that AMPK would promote ferroptosis by inactivating mTORC1. However, under AMPK activation conditions, the suppression of PUFA biosynthesis presumably would override any pro-ferroptosis effect downstream of AMPK, resulting in ferroptosis inhibition (i.e., without appropriate PUFA biosynthesis, ferroptosis cannot

ensue regardless of whether cells maintain intact cellular defense systems for lipid peroxidation detoxification or not).

The Rag GTPases play important roles in mTORC1 localization on lysosomes and its subsequent activation in response to amino-acid stimulation[35]. Our data showed that cyst(e)ine regulation of mTORC1 activation and GPX4 levels is independent of

**Fig. 4 mTORC1 inhibition sensitizes cancer cells or tumors to ferroptosis. a** UMRC6 cells were treated with 1 μM RSL3 (or 1 μM ML162), or 1 μM Torin1, or both combined with or without 5 μM Ferrostatin-1 (Ferr-1) or 100 μM deferoxamine (DFO) for 6 h. Then lipid peroxidation was assessed using BODIPY™ 581/591 C11 staining followed by FACS analysis. **b** Bar graph showing the viability of UMRC6 cells treated with drugs using indicated combination. Concentration of each drug is described as follows. RSL3, 1 μM; ML162, 1 μM; Torin1, 1 μM; Ferr-1, 5 μM; DFO, 100 μM. $n = 3$. **c** Bar graph showing the viability of indicated cells treated with 300 nM RSL3, or 1 μM Torin1, or both for 8 h. $n = 5$. **d** Bar graph showing the viability of indicated cells treated with 300 nM ML162, or 1 μM Torin1, or both for 8 h. $n = 5$. **e** Bar graph showing the viability of indicated cells treated with different concentrations of cystine for 48 h. $n = 4$. **f** Plot showing the viability of indicated cells treated with different concentrations of erastin for 36 h. $n = 4$. **g** Volumes of PDX tumors in mice treated daily with 30 mg/kg IKE, or 10 mg/kg AZD8055, or both at different time points as shown. $n = 5$ mice per group. **h** Percentages of P-4EBP1-, GPX4-, or 4-HNE-positive stained cells per field. $n = 5$ randomly selected high-power fields per group. **i** Haematoxylin and eosin and immunohistochemical staining of PDX tumor samples collected from mice at the end of treatments described in **g**. For all panels, error bars are mean ± SD. If not otherwise specified, n indicates biologically independent repeats. *P* value was determined by two-tailed unpaired Student's *t* test. For **f** and **g**, *p* value was determined by two-way ANOVA test. Source data are provided as a Source Data file.

TSC but depends on Rag, which is consistent with a recent study showing that cyst(e)ine activates mTORC1 through Rags[50]. We noticed that cystine-induced mTORC1 activation is not completely abolished in our *RagA/B* KO cells (Fig. 3f); therefore, there might exist additional mechanisms mediating cystine-induced mTORC1 activation, which is consistent with previous findings, demonstrating Rag-independent but Arf-1- or Rab-1A-dependent mechanisms to mediate mTORC1 activation in response to some amino acids[51,52]. Further studies are required to understand the exact mechanisms by which cyst(e)ine regulates mTORC1 signaling, including identifying potential cyst(e)ine sensors for mTORC1 activation.

Our data showed that *RagA/B* deletion in UMRC6 cells does not apparently reduce mTORC1 signaling under normal culture conditions (Fig. 3e). Previous studies showed that acute deletion of *RagA* in MEFs significantly decreased basal mTORC1 signaling[53], but MEFs with stable RagA/B deletion only exhibited very moderate reduction in mTORC1 signaling under normal culture conditions[51]. *RagA/B* deletion in the heart even slightly increased mTORC1 signaling[54]. RagA is essential for cell growth in MEFs[53]. Likewise, we also noticed that it is challenging to generate *RagA/B* complete KO UMRC6 cells from single clones using CRISPR approaches. In our study, we have used pooled CRISPR *RagA/B* KO cells, which maintain residual RagA and B expression (Fig. 3e). It is possible that residual RagA/B expression is required to maintain mTORC1 activation and cell growth under basal conditions, and/or RagA/B deficiency induces compensatory mTORC1 activation in order to allow for long-term culture of these *RagA/B* KO cells, which likely explains the lack of obvious reduction in basal mTORC1 signaling in *RagA/B* KO cells generated in our and some other studies. Apparently, this residual RagA/B expression is not sufficient to maintain cystine-induced mTORC1 activation, resulting in a significant attenuation of cystine-induced mTORC1 activation in *RagA/B* KO cells.

We noticed that cystine starvation, but not the deprivation of some other amino acids such as leucine, significantly decreased GPX4 protein levels (Supplementary Fig. 8). It is possible that, although both leucine and cystine starvation decrease GPX4 protein synthesis by inactivating mTORC1, leucine starvation (but not cystine starvation) might also induce adaptive responses to restore GPX4 protein level, resulting in differential GPX4 protein level change upon cystine or leucine starvation. Alternatively, scaffold mechanisms might facilitate to link cyst(e)ine stimulation to specific downstream effectors of mTORC1. A recent study revealed that such mechanisms underlie the differential regulation of TFEB by amino acids and serum through mTORC1[55]. It is also possible that starvation of different amino acids affects mTORC1 signaling quantitatively differently, resulting in differential regulation of downstream targets in protein synthesis (however, such quantitative differences might

not be readily revealed by S6K or 4EBP phosphorylation). Further studies will be directed to understand these interesting questions.

Our fundamental understanding of cyst(e)ine and mTORC1 regulation of GPX4 protein synthesis also has important implications for cancer therapies. A recent study showed that systemic depletion of cyst(e)ine with cyst(e)inase in vivo significantly suppresses mTORC1 signaling[56], which is line with our data that cystine starvation potently inhibits mTORC1 signaling. We further showed that, although mTORC1 inhibition is not sufficient to induce ferroptosis (likely because GPX4 protein synthesis is reduced but not totally abolished upon mTORC1 inhibition), mTORC1 inhibitors significantly sensitizes cancer cells to ferroptosis induced by FINs. Our preclinical studies using PDXs also showed that combined treatment with mTORC1 inhibitor and FIN synergistically depleted GPX4 and induced ferroptosis in vivo, resulting in more potent tumor growth suppression than either treatment alone. It should be noted that in our study we chose the dose of AZD8055 (10 mg/kg) that was used in the original publication on AZD8055[34] as well as multiple other studies. While this dose effectively suppresses mTORC1 in tumors in some other studies, it only moderately inhibits mTORC1 in the PDX model used in our study. This discrepancy is likely caused by tumor heterogeneity across different tumor models. Because mTORC1 inhibition generally induces limited cytotoxic effects, adding FINs can boost the cytotoxic effect by inducing ferroptosis and enhance the therapeutic efficacy. Our studies, therefore, call for further exploration of the combination of mTORC1 inhibitors and FINs in cancer treatment.

## Methods

**Cell culture studies**. UMRC6 cell line was purchased from Sigma (#08090513). HEK293T (CRL-3216), 786-0 (CRL-1932), ACHN (CRL-1611), NCI-H226 (CRL-5826), H460(HTB-177), NCI-H23(CRL-5800), NCI-H1299(CRL-5803) cell lines were obtained from ATCC. The work does not involve any cancer cell lines that are listed in the database of commonly misidentified cell lines. Wild-type and *TSC1* KO MEFs were generated as described in our previous publication[57]. UMRC6 and MEFs were cultured in Dulbecco's Modified Eagle Medium (DMEM) containing penicillin (100 units/mL), streptomycin (100 μg/mL), and 10% (vol/vol) fetal bovine serum (FBS). NCI-H226, NCI-H1299, NCI-H23, 786-O, and H460 cells were cultured in RPMI-1640 medium. ACHN cells was cultured in ATCC-formulated Eagle's Minimum Essential Medium. For the cystine deprivation treatment, cells were cultured in cystine-free DMEM with different concentrations of cystine + 10% (vol/vol) dialyzed FBS as previously described[24]. Serine-free medium was prepared based on the media purchased from US biological (D9802-01). Leucine-free medium was prepared based on the media purchased from ThermoFisher Scientific (30030). Stable cell lines were generated as described in our previous publication[24]. In brief, the plasmid used for knockout or over-expression of target gene was co-transfected with lentivirus packaging vector psPAX.2 and envelope plasmid pMD2.G into HEK293T cells for 48 h. Then, culture medium of transfected cells was collected and filtered to infect target cell line in the presence of 8 μg/ml polybrene (Millipore, TR-1003) for 6–12 h. Then fresh medium was added to culture cells for additional 24 h followed by selection with appropriate antibiotics.

**Constructs and reagents**. SLC7A11-KO, LAMP2 KO, 4EBP1/2 DKO, and RagA/B DKO cell lines were generated using CRISPR/Cas9 technology as previously described[58]. In brief, the sgRNAs were cloned into the lentiviral lentiCRISPR v2 vector. All constructs were confirmed by DNA sequencing. pGIPZ-shRNAs against GPX4 were obtained from the Functional Genomics Core at The University of Texas MD Anderson Cancer Center. The sequences of the primers used in PCR mutagenesis, gRNAs, and shRNAs used in this study are listed in Supplementary Table 3. pLenti-SLC7A11-V5 construct was generated as described in our previous publications[24,59]. pcDNA-flag-GPX4 expression plasmid was a gift from Dr. Aikseng Ooi at The University of Arizona Health Sciences[60]. pCW57.1-4EBP1 4 A construct was obtained from Addgene (#38240). Other reagents were purchased as follows: Erastin (E7781), ferrostatin-1 (SML0583), deferoxamine mesylate salt (DFO, D9533), GSHEE (G1404), MG132 (M7449), chloroquine diphosphate (C6628), and doxycycline (D9891) were obtained from Sigma. Torin1 was obtained from ApexBio (A8312). AZD8055 (S1555) and Salubrinal (S2923) were obtained from Selleckchem. IKE was obtained from MedchemExpress (HY-114481).

**Mass spectrometry analysis**. Cells in triplicate with or without treatment were lysed in chilled lysis buffer (8.0 M urea in 0.1 M NH₄HCO₃, supplemented with 1× protease inhibitor cocktail). Equal proteins from each sample were reduced with 5 mM dithiothreitol, alkylated with 15 mM iodoacetamide, and then quenched by 15 mM dithiothreitol. Samples were sequentially digested by Lys-C (enzyme: proteins, 1:100) for 4 h and then trypsin (enzyme: proteins, 1:50) overnight at 37 °C. The digestion was quenched by trifluoroacetic acid to a final 0.1% concentration. The tryptic peptides were pre-separated by high-pH reverse-phase HPLC with a Waters XBridge C18 column (3.7-μm particles, 4.6 × 100 mm). Eluents were collected every 1 min in a 30-min gradient from 2% to 80% of buffer B (2% H₂O, 98% ACN; buffer A: 98% H₂O, 2% ACN, NH₄OH, pH 10.5) at a flow rate of 0.6 ml/min. Si fractions were pooled from collected eluents using a previously reported method[61]. Fractions from the above fractionation were dissolved with solvent A (0.1% formic acid in H₂O) and separated with a homemade reversed-phase 25-cm column (75 μm ID, 1.9 μm C18) in a 65-min gradient from 5% to 50% solvent B (0.1% formic acid in 80% ACN) using the EASY-nLC 1200 system (Thermo Fisher Scientific). The eluted peptides were ionized and introduced into a Q Exactive HF-X mass spectrometer (Thermo Fisher Scientific). The precursor scan was set as 375-1500 m/z with a resolution of 70,000 at m/z 200. Precursor ions with one charge or five or more charges were excluded. The maximum injection time was 100 ms and the automatic gain control target 1e6 for precursor ions. The 40 most intense ions above 1.5e4 were isolated and sequentially fragmentized by higher collision dissociation with normalized collision energy of 28%, with 1 m/z isolation windows. Ion fragments were detected in the Orbitrap at a resolution of 17,500 at m/z 200 with an automatic gain control 1e6 and 100 ms of maximum injection time. The acquired MS/MS raw data were processed using MaxQuant software (version 1.6.5.0) and searched against the human proteomes database from uniprot (26 January 2019, updated) with a reversed decoy database by Andromeda search engine. Label-free quantification (LFQ) was set with a default parameter. The proteingroup.txt file was used to next analysis with Perseus (version 1.6.7.0)[62]. The principal component analysis was based on the LFQ intensities of all data. To identify significantly modulated proteins across starvation and control, we performed a Student's t test for identified targets. To correct generated p value for multiple testing, we calculated q value with the parameters a permutation-based false discovery rate < 0.05 and S0 = 0.1 by the Perseus software[62].

**Cell death/viability assay**. Cell viability or cell death was measured as described previously[63–65]. To measure cell viability, 10,000 cells per well were seeded in 96-well plates and treated as indicated, after which the medium in each well was replaced with 100 μl fresh medium containing 10% Cell Counting Kit-8 reagent (APExBIO, K1018). After incubation for 1 h at 37 °C, plate was read by a FLUOstar Omega microplate reader (BMG Labtech) at an absorbance of 450 nm. Cell viability (%) = [(Absorbance of tested compound minus Absorbance of blank)/ (Absorbance of control minus Absorbance of blank)] × 100. To measure cell death, cells were seeded in 12-well plates and treated as indicated. Then cells were collected in a 1.5-mL tube, washed once with phosphate-buffered saline (PBS) and stained with 1 μg/mL propidium iodide (PI) (Roche) in PBS. Dead (PI-positive) cells were detected by a BD Accuri C6 flow cytometer (BD Biosciences). The gating strategy used for PI staining analysis was shown in Supplementary Figure 9.

**Light microscopy and immunofluorescence microscopy**. For light microscopy, cells cultured in 6-well or 12-well plates were treated as indicated in the figure legends. Phase-contrast images were obtained with an EVOSfl (Advanced Microscopy Group) microscope. Immunofluorescence microscopy was conducted as previously described[66]. Cells were cultured on glass coverslips and washed with PBS, then fixed in 3.7% formaldehyde. After permeabilization for 10 min in 0.1% TritonX-100/PBS and incubation with blocking buffer (5% bovine serum albumin in 0.1% Triton/PBS), cells were by incubated with mTOR (1:500; Cell Signaling Technology, #2983) and LAMP2 (1:500; Abcam, 25631) antibodies overnight. Cells were washed with PBS and incubated with fluorescent secondary antibodies (ThermoFisher Scientific) for 2 h. The nuclei were labeled with 4′,6-diamidino-2-

phenylindole, ThermoFisher Scientific), and fluorescent image was captured using a confocal microscope (Leica).

**Real-time PCR**. Real-time PCR was done as previously described[67,68]. Total mRNA was extracted from samples using TRIzol reagent (ThermoFisher Scientific) according to the manufacturer's instructions. Then, 2 μg of RNA was used to generated cDNA through SuperScript II reverse transcriptase (ThermoFisher Scientific). Real-time PCR was conducted in triplicate in a 20-μL reaction mixture by using SYBR GreenER qPCR SuperMix Universal (ThermoFisher Scientific, #11762100). β-actin was used as internal control. The primer sequences used are listed in Supplementary Table 3.

**Western blotting**. Protein levels were determined by immunoblotting as previously described[69,70]. Specifically, 15 μg protein per sample was loaded and subjected to electrophoresis for 1–2 h to separate the target protein. After transferring the protein to a polyvinylidene difluoride membrane (Biorad, 1620177) and blocking for 1 h using 5% solution of nonfat powdered milk in Tris-buffered saline, the membrane was incubated with appropriate primary antibody at 4 °C overnight with gently shaking. Then protein of interest was observed through HRP-conjugated anti-rabbit or -mouse secondary antibody (Invitrogen, 1:5000, G-21040/ G-21234) and Pierce™ ECL Plus Western Blotting Substrate (ThermoFisher Scientific, 32132). The original records of protein blots were included in Supplementary Figure 10. The primary antibodies and concentrations used for Western blotting were: GXP4 (1:1000; R&D systems, MAB5457), tubulin (1:5000; Cell Signaling Technology, #2144), vinculin (1:10000; Sigma, V4505), SLC7A11 (1:4000; Cell Signaling Technology, #12691) p53 (1:1000; Santa Cruze, sc-126), LC3B (1:5000; Cell Signaling Technology, #3868), HIF2α (1:1000; Cell Signaling Technology, #7096), TSC1 (1:1000; Cell Signaling Technology, #6935) phospho-S6K (1:1000; Cell Signaling Technology, #9205), S6K (1:1000; Cell Signaling Technology, #9202), P-S6 (1:1000; Cell Signaling Technology, #2215), S6 (1:1000; Cell Signaling Technology, #2217), P-4EBP1 (1:1000; Cell Signaling Technology, #2855), 4EBP1 (1:1000; Cell Signaling Technology, #9644), 4EBP2 (1:1000; Cell Signaling Technology, #2845), RagA (1:1000; Cell Signaling Technology, #4357), RagB (1:1000; Cell Signaling Technology, #8150), LAMP2 (1:1000; Cell Signaling Technology, #49067), P-eIF2α (1:1000; Cell Signaling Technology, #3398), eIF2α (1:1000; Cell Signaling Technology, #5324), Actin (1:1000; Cell Signaling Technology, #3700), Rictor (1:1000; Cell Signaling Technology, #2140).

**Cycloheximide chase analysis**. To determine GPX4 protein stability, UMRC6 cells were cultured in media with or without cystine for 24 h. Then, cycloheximide (50 μg/ml) were added to media and cells were collected at different time points on ice. Protein level for each sample was analyzed by Western blotting and quantified using normalized ratio of GPX4 to tubulin.

**Lipid peroxidation assay**. To measure levels of lipid peroxidation, cells in 12-well plate after treatments were incubated with fresh medium containing 2 μM BODIPY 581/591 C11 dye (Invitrogen, D3861) for 30 min. Then cells were collected and washed once with PBS followed by fluorescence-activated cell sorting (FACS) analysis. Fluorescence in channel 1 in live cells was captured and plotted using FlowJo_V10 software. The gating strategy used for the assay was shown in Supplementary Figure 9.

**Cystine uptake assay**. Cells were seeded in 12-well plates and treated as described for individual experiment. To measure cystine uptake levels of cells upon treatment, each well was replaced with cystine uptake medium containing 1 μM cystine and 0.04 μCi 2-[1-14 C] labeled cystine (PekinElmer). Then plate was incubated for indicated time followed by washing with PBS and lysing in 0.1 mM NaOH. Radioactivity of intracellular labeled cystine was measured with a Tri-Carb Liquid Scintillation Analyzer (PerKinElmer, Model 4810TR) according to the manufacturer's instructions. All experiments were conducted with three independent replicates.

**Luciferase reporter assay**. The GPX4-3′-UTR sequence was amplified from pcDNA-flag-GPX4 plasmid and cloned into pGL3 luciferase reporter vectors. Primers used to clone 3′-UTR of GPX4 gene are listed in Supplementary Table 3. Luciferase reporter assay was conducted using a Dual-Luciferase Reporter Assay System (Promega, E1910) according to the manufacturer's instructions. In Brief, UMRC6 cells were transfected with the appropriate plasmids for 24 h and split into 96-well plate followed by treatment as described in the legends. Then, cells were washed with PBS, and lysed for 15 min at room temperature. Cell lysate was added to a white opaque 96-well plate for subsequent luciferase activity measurement. Luminescence from at least four independent samples was recorded using a Gen5 microplate reader (BIOTEK).

**Polysome profiling analysis**. Sucrose solutions were prepared in polysome buffer (10 mM HEPES, pH 7.4, 100 mM KCl, 5 mM MgCl2 and 100 mg/ml cycloheximide). A 15–45% (w/v) sucrose density gradient was freshly prepared in a SW41 ultracentrifuge tube (Backman) using a Gradient Master (BioComp Instruments).

Cells were lysed in polysome lysis buffer (polysome buffer and 2% TritonX-100) and cell debris were removed by centrifugation at $13,000 \times g$ for 10 min at 4 °C. In all, 500 mL of supernatant was loaded onto sucrose gradients followed by centrifugation for 2 h 30 min at $36,000 \times g$ and 4 °C in a SW41 rotor. Separated samples were fractionated at 0.75 ml/min through an automated fractionation system (Isco) that continually monitors OD254 values. An aliquot of each ribosome fraction were spiked with isolated luciferase RNA and then used to extract total RNA using Trizol LS reagent (Invitrogen) for RT-qPCR.

**Protein synthesis assays.** UMRC6 cells were plated in 35 mm dishes. After 12 h of culture in DMEM (+10% FBS), cells were rinsed with PBS and supplemented with either cystine-free media (+10% DFBS) for 22 h or in DMEM (+10% dialyzed FBS) with 10 μM erastin for 24 h. In all, 10 μM puromycin was added to the medium, and cells were harvested 10 min after the addition of puromycin. Cells were washed twice with ice-cold PBS and were then lysed in 100 μL of cell lysis buffer (50 mM Tris, pH 7.5, 150 mM NaCl, 1 mM EDTA, 0.5% v/v Nonidet NP-40) supplemented with Complete protease inhibitor and PhosSTOP phosphatase inhibitor (Roche). Cell lysate was mixed with an equal volume of 2× sodium dodecyl sulfate polyacrylamide gel electrophoresis (SDS-PAGE) loading buffer (0.1 M Tris-HCl, pH 6.8, 4% w/v SDS, 20% v/v glycerol, 200 mM dithiothreitol, and 0.05% w/v bromophenol blue). Proteins were separated by SDS-PAGE (10% w/v polyacrylamide) and were then transferred to Immobilon-P membranes. Membranes were blocked for 1 h in Tris-buffered saline (50 mM Tris, 150 mM NaCl, pH 7.6) containing 5% w/v nonfat milk and 0.1% v/v Tween-20. Puromycin-labeled polypeptides were then quantified by incubating membranes with puromycin antibody (Developmental Studies Hybridoma Bank, 1:1000, #PMY-2A4) overnight at 4 °C and then with horseradish peroxidase-coupled secondary antibodies at room temperature for 1 h. Immunoblots were visualized using enhanced chemiluminescence and β-Actin antibody (Sigma-Aldrich, 1:5000, A2228) was used to quantify β-actin as a loading control.

**PDX experiments.** PDXs were generated in accordance with protocols approved by the Institutional Review Board at The University of Texas MD Anderson Cancer Center. Informed consent was obtained from the patients and the study is compliant with all relevant ethical regulations regarding research involving human participants. Xenograft experiments were performed in accordance with a protocol approved by the Institutional Animal Care and Use Committee and Institutional Review Board at The University of Texas MD Anderson Cancer Center. All the NOD-scid gamma (NSG) mice were purchased from the Experimental Radiation Oncology Breeding Core Facility at MD Anderson Cancer Center and housed in the Animal Care Facility at the Department of Veterinary Medicine and Surgery at MD Anderson Cancer Center. Mice were maintained at a condition of 12-h light/ 12-h dark cycle and temperatures of 65–75 °F (~18–23 °C) with 40–60% humidity. PDX experiments were performed as previously described[71]. Specifically, PDX tumor derived from lung cancer patient rinsed in cold DMEM media were minced into fragments 1–2 mm$^3$ in volume. Then tumor fragment was subcutaneously inoculated into the dorsal flank of NSG mice. The tumor growth in mice was monitored by bi-dimensional tumor measurements. Tumor volume was calculated according to the equation $v = \text{length} * \text{width}^2 * 1/2$. When tumors grew to a volume of 200 mm$^3$, the mice were divided randomly into four groups ($n = 5/$ group) and treated with vehicle, 10 mg/kg AZD8055, 30 mg/kg IKE, or both (10% dimethyl sulfoxide/90% corn oil) by daily intraperitoneal administration. Body weights of mice in each group during treatment were also recorded accordingly.

**Histology and immunohistochemistry.** Xenograft tissue samples were collected at the end of treatment and immediately fixed in 10% neutral-buffered formalin (ThermoFisher Scientific) for at least 6 h. Then tumor samples were subjected to embedding, section and hematoxylin and eosin staining. For immunohistochemistry staining, tissue sections were processed according to methods described in our previous publications[72,73]. Specifically, the paraffin-embedded tissue section after deparaffinization and rehydration was subject to antigen retrieval in citrate-based unmasking solution (Vector Laboratories, H-3300-250) in a steam pot for 35 min. Next, the slide was washed by PBS and quenched in the buffer containing 10% methanol and 0.1% $H_2O_2$ for 30 min. After blocking in goat serum for 1 h at room temperature, specific antibody of interest was added and incubated overnight at 4 °C, followed by reaction with a biotinylated secondary antibody and detection using a commercial ABC-HRP Kit (Vector Laboratories, PK-4000). Tissue microarray of kidney cancer was purchased from US Biomax (BC07115a) followed by immunohistochemistry staining with indicated antibodies. The antibodies used for immunohistochemistry staining were anti-GPX4 (1:100, Novus Biologicals, NBP2-54979), P-4EBP1 (1:100, Cell Signaling Technology, #2855) anti-4-HNE (1:200, Abcam, ab46545). Representative images were obtained at ×200 magnification using a microscope (Olympus, BX43).

**Statistics and reproducibility.** For all statistical analyses, the difference was considered significant with a $p$ value <0.05. Comparisons between two conditions or groups were analyzed by two-tailed Student's $t$ tests in GraphPad Prism (GraphPad Software, Inc.). Two-way analysis of variance was used to calculate differences between two curves with multiple time or concentration points. Data

are presented as mean ± S.D., with at least three biologically independent replicates in each group. The detailed statistic for each plot was described in figure legends. For immunoblots and treatment assays, the experiments have been repeated at least twice with similar results and representative data was shown.

**Reporting summary.** Further information on research design is available in the Nature Research Reporting Summary linked to this article.

## Data availability
The raw data and processed data for mass spectrometry analysis of proteins in UMRC6 cells upon cystine deprivation have been deposited to MassIVE data sets with the identifier MSV000086009. Uniprot is a public and freely accessible resource of protein sequence and functional information (https://www.uniprot.org/). The uncropped films for immunoblots used in this study have been shown in Supplementary Figure 10. All other data that support the findings of this study are available from the corresponding author upon reasonable request. Source data are provided with this paper.

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

## Acknowledgements

This research has been supported by Institutional Research Fund from The University of Texas MD Anderson Cancer Center, KC180131 from Department of Defense Kidney Cancer Research Program, R01CA181196, R01CA244144, and R01CA247992 from the National Institutes of Health (to B.G.). B.G. is an Andrew Sabin Family Fellow. P.K. is supported by CPRIT Research Training Grant (RP170067) and Dr. John J. Kopchick Research Award from The University of Texas MD Anderson Cancer Center UTHealth Graduate School of Biomedical Sciences. Work in the lab of S.-B.Q. is supported by NIH grants R01GM1222814 and R21CA227917 and by the Howard Hughes Medical Institute (55108556). R. S. is supported by an NIH, National Institute of General Medical Sciences, Chemical Biology Interface (CBI) Training Grant (T32GM008500). PDX generation and annotation were supported by the University of Texas MD Anderson Cancer Center Moon Shots Program, Specialized Program of Research Excellence (SPORE) grant CA070907, and University of Texas PDX Development and Trial Center grant U54CA224065. This research has also been supported by the National Institutes of Health Cancer Center Support Grant P30CA016672 to The University of Texas MD Anderson Cancer Center.

## Author contributions

Y.Z. performed most of the experiments with assistance from X.L., H.L., C.M., G.L., P.K., W.C., J.Z., Z.X., and L.Z.; R.S. performed polysome analysis under the guidance of S.-B.Q.; L.N., and C.W. performed proteomic analyses under the guidance of J.C.; B.F. provided PDX models; B.G. designed the experiments, supervised the study, and provided funding support; B.G. wrote most of the manuscript with assistance from other co-authors; all authors commented on the manuscript.

## Competing interests

The authors declare no competing interest.
