## [Peer Review File · Nature Communications]

REVIEWER COMMENTS

Reviewer #1 (Remarks to the Author):

In this manuscript, Zhang et al. propose that SLC7A11-mediated cyst(e)ine uptake promotes the protein synthesis of the central ferroptosis inhibitor, GPx4, and that this process is mediated by mTORC1. Their data indicate that cysteine deprivation inhibits mTORC1 and that mTORC1 mediates GPx4 translation. Overall, the authors present a strong argument and the study has clear cancer therapeutic relevance. There are several issues underlying the major mechanistic conclusion and therapeutic potential of the study that the authors should address with additional experiments and more thorough discussion:

Major points

1. The authors show that CQ and MG132 do not block the decrease in GPx4 following cystine deprivation. However, it is possible that autophagy and proteasomal degradation can compensate for one another to degrade GPx4, which will be highly oxidized due to the lack of GSH. It would be worthwhile to see if blocking both degradation pathways can maintain GPx4 protein levels.
2. It is also important to test the involvement of LAMP2a and chaperone-mediated autophagy in the degradation of GPx4 upon erastin treatment, a mechanism reported previously (PNAS 2019, 116:2996) and obviously contradictory with the current study. This is an issue the authors should resolve by conducting LAMP2a knock-out study.
3. In Figure 2H-J, the authors go into the surprising relationship between SLC7A11 and sensitivity to GPx4 inhibitors. However, the possibility remains that increased thiol abundance could be just enough to block cell death under certain circumstances. One way to demonstrate that this is not the case would be to overexpress or knockdown SLC7A11 in a GPx4 knock-out background, cultured with an antioxidant. If there is no change in the rate of cell death (upon antioxidant withdrawal) then the authors' hypothesis stands.
4. The authors find that BSO, unlike erastin or cystine deprivation, cannot decrease GPx4 expression. However, based on their later experiments, BSO should be more potent as a ferroptosis inducer with mTOR inhibition. Is this the case? Also, control experiments such as how BSO affects mTORC1 activity should be included.
5. While the lack of class II FINs for animal studies is a problem, using IKE is even more problematic in the context of this manuscript. Combining AZD8055, a potent antitumor drug in animals, with IKE, a potent SLC7A11 inhibitor, makes the PDX experiment unconvincing in terms of advancing the authors' argument (as IKE alone, even in the absence of AZD8055, should both deplete GSH and inhibit mTORC1, according to the mechanism proposed by the authors), although the combination therapy does show better effect than either single treatment. To directly support the hypothesis that mTOR inhibition can sensitize tumors to GPX4 inactivation, the authors have one good option: conduct the PDX experiment using BSO instead of IKE, +/- AZD. That is why the previous point is so critical.
6. Does the withdrawal of other amino acids required for sustaining mTORC1 activity (such as leucine) also decrease GPx4 expression and enhance ferroptosis induced by class II FINs, as predicted by author's mechanism? Positive data will be a strong evidence supporting the proposed mechanism.
7. The tissue microarray in Figure 3j-k is unconvincing without another control protein, since 4EBP1 is a general regulator of translation.

Minor points:

- In Figure 3A, the authors show the effect of cystine deprivation on mTOR targets. They are missing a panel for p-4EBP1; although the phosphorylation levels can reasonably be assumed from the size shift in 4EBP1, it may not be immediately clear to those unfamiliar with the protein, especially since p-4EBP1 is in the adjacent panels.
- In Figure 4A (left) there appear to be gates on the flow cytometry charts that do not correspond to the percentages shown.

Reviewer #2 (Remarks to the Author):

In this manuscript, Zhang et al. describes an unexpected role of cyst(e)ine mediated mTORC1 activation in regulating GPX4 protein synthesis. The authors showed that cyst(e)ine depletion inhibits the engagement of GPX4 mRNA with polysomes, and nicely ruled out the possibility that Cys or mTORC1 is regulating GPX4 protein degradation. The alterations in GPX4 protein levels have direct consequences in the cellular sensitivity to ferroptosis, and authors revealed that co-inhibition of mTORC1 and GPX4 has synergistic effects in ferroptosis induction.

I am reading this paper with great interest, and I found the findings described here are novel and significant in the following ways:

1. Although the central role of GPX4 in safe-guarding cells from ferroptosis is well-appreciated, the regulation of GPX4 protein levels is poorly understood and largely overlooked. Previous reports in Zou et al., Nature Communications 2019 suggested that the GPX4 transcript levels are largely constant across more than 900 cancer cell lines from various lineages. This work is consistent with Zou et al. in showing that GPX4 mRNA levels are not changed by the chemical and genetic perturbations introduced here, and further describes the important findings on how cyst(e)ine availability is rate-limiting in GPX4 protein synthesis.

2. The exact mechanism of erastin-induced ferroptosis has not been fully resolved. This work suggests that in addition to blocking glutathione synthesis (thus reducing the abundances of the GPX4 co-factor), cyst(e)ine-depletion also inhibits GPX4 protein synthesis, suggesting a novel mechanism underlying the activity of erastin.

3. In this referee's lab, we have anecdotally noticed that mTORC1 inhibition sensitizes several cancer cell lines to GPX4 inhibition-induced ferroptosis, as well as the decreased GPX4 protein levels in response to erastin treatment in cancer cells. We were intrigued by these observations but did not understand the molecular mechanism underlying this connection. In this referee's opinion, the current study provides a very convincing explanation to the crosstalk between mTORC1 signaling and GPX4 activity.

4. The combinatorial strategy of ferroptosis induction coupled with mTORC1 inhibition is very appealing from a therapeutic perspective for augmenting the efficacy of ferroptosis-inducing approaches.

Overall, this study is performed with high rigor and the conclusions are largely supported by well-controlled experimental data. The manuscript is well-written and easy to understand. The conclusions in this study is expected have direct impact in the fields of ferroptosis, amino acid metabolism and mTORC1 signaling. It is appreciated that the authors validated each of their observations in multiple cell line models, and used cDNAs to rescue most of the genetic perturbations introduced in this study. Despite that how Cys regulates mTORC1 activity remains an intriguing scientific gap, this manuscript is complete on its own. This referee does not have additional experimental requests other than the minor suggestions listed below:

1. The authors may briefly discuss (in the Discussion) their work in light of the recent reports in [Liu et al., Cancer Gene Therapy 2020 Interplay between MTOR and GPX4 signaling modulates autophagy dependent ferroptotic cancer cell death], and [Reinke et al., Plos ONE, 2014 Translational Regulation of GPx-1 and GPx-4 by the mTOR Pathway].

Minor comments:

Figure 1a, the authors should specify whether multi-testing correction is performed with the p values presented in the volcano plot. If raw p value was presented here, this referee recommends using

multiple-testing corrected data.

Figure 1n, the label of the y-axis has overlapped text

Figure 2e, scale bar of the images is missing.

Figure 4a, there are several typos with regard to Torin1

Figure S4b, scale bar labeling, 100 uM should be 100 um.

Line 228, typo in "Torrin1"

Reviewer #3 (Remarks to the Author):

In this manuscript, authors investigated mechanisms that regulate GPX4, an inhibitor of ferroptosis. The results show that cystine activates mTORC1 and promotes cap-dependent translation of GPX4 mRNA. Pharmacologic inhibition of mTORC1 decreases GPX4 protein synthesis and sensitizes cancer cells to ferroptosis. Overall, the studies are interesting that provide a possible mechanistic link between nutrient signaling and ferroptosis. However, some key findings are still preliminary that need to be strengthened before the manuscript is suitable for publication.

1. Authors concluded that cystine activates mTORC1 through a Rag-dependent mechanism. However, the data are largely correlative. Authors should demonstrate that cystine regulates interaction of Rag with mTORC1 and that activation of mTORC1 by cystine is Rag-dependent.

2. Authors showed that treatment with Torin1 and AZD8055, not rapamycin decreases GPX4 protein level. Since Torin1 and AZD8055 are mTOR kinase inhibitors that target both mTORC1 and mTORC2, this observation suggests that mTORC2 is involved. Authors should address if mTORC2 plays a role in GPX4 regulation.

3. It was previously shown that cystine regulates mTOR signaling through eIF2alpha (Sci Rep 6: 30033). Authors should clarify whether eIF2alpha is involved and how it is related to the proposed mechanism in the present study.

Detailed Point-by-point response to the reviewer's comments:

Note to reviewers: We thank the reviewers for taking efforts to review our manuscript and for providing insightful comments to further improve our manuscript. Below we provide the detailed point-by-point response to address all the comments raised by the reviewers. To facilitate the review of our rebuttal letter and manuscript by reviewers, we present all the new data as rebuttal letter figures in this letter, with referrals to corresponding figures and text in our revised manuscript. We have also marked all the changes in our revised manuscript by colored text.

Reviewer #1 (Reviewer Comments to the Author):

In this manuscript, Zhang et al. propose that SLC7A11-mediated cyst(e)ine uptake promotes the protein synthesis of the central ferroptosis inhibitor, GPx4, and that this process is mediated by mTORC1. Their data indicate that cysteine deprivation inhibits mTORC1 and that mTORC1 mediates GPx4 translation. Overall, the authors present a strong argument and the study has clear cancer therapeutic relevance. There are several issues underlying the major mechanistic conclusion and therapeutic potential of the study that the authors should address with additional experiments and more thorough discussion:

We appreciate the positive and insightful comments from this reviewer. We hope that our revision now has addressed the critiques from this reviewer.

Major points

1. The authors show that CQ and MG132 do not block the decrease in GPx4 following cystine deprivation. However, it is possible that autophagy and proteasomal degradation can compensate for one another to degrade GPx4, which will be highly oxidized due to the lack of GSH. It would be worthwhile to see if blocking both degradation pathways can maintain GPx4 protein levels.

The reviewer provided a nice suggestion to help us further clarify the role of protein degradation in GPX4 regulation by cystine starvation. As suggested, we treated cells with both chloroquine (CQ) and MG132 under cystine starvation or erastin treatment condition. As shown in **Rebuttal Figure 1** (Figure S1q in our revised manuscript), cystine starvation or erastin treatment decreased GPX4 protein level; however, combined treatment with CQ and MG132 did not obviously restore GPX4 protein level under cystine starvation or erastin treatment. As a control, we showed that combined treatment with CQ and MG132 increased both HIF2 α and LC3B levels. These data further support our finding that cystine starvation regulates GPX4 protein levels in a protein degradation-independent manner.

Rebuttal Figure 1. The effect of CQ and MG132 on GPX4 protein regulation upon erastin treatment or cystine deprivation. a, UMR6 cells were treated with 10 μ M erastin with or without 25 μ M CQ and 5 μ M MG132 for 16 hours followed by Western blotting analysis of indicated proteins. b, UMR6 cells were cultured in cystine-free media with or without 25 μ M CQ and 5 μ M MG132 for 16 hours followed by Western blotting analysis of indicated proteins.

2. It is also important to test the involvement of LAMP2a and chaperone-mediated autophagy in the degradation of GPx4 upon erastin treatment, a mechanism reported previously (PNAS 2019, 116:2996) and obviously contradictory with the current study. This is an issue the authors should resolve by conducting LAMP2a knock-out study.

We conducted this experiment as the reviewer kindly suggested. As shown in **Rebuttal Figure 2** (Figure S1r in our revised manuscript), LAMP2 knockout in UMRC6 cells did not affect the reduction of GPX4 protein level under erastin treatment or cystine starvation, suggesting that, at least in the cell line we have examined, cystine starvation regulates GPX4 protein levels independent of LAMP2 or chaperone-mediated autophagy.

Rebuttal Figure 2. The effect of LAMP2 deletion on GPX4 protein regulation upon erastin treatment or cystine deprivation. a, UMRC6 control (sgC) and LAMP2-knockout (sgLAMP2) cells were treated with erastin with indicated concentrations (μM) for 24 hours followed by Western blotting analysis b, UMRC6 sgC and sgLAMP2 cells were cultured in media with or without cystine for 24 hours followed by Western blotting analysis.

3. In Figure 2H-J, the authors go into the surprising relationship between SLC7A11 and sensitivity to GPx4 inhibitors. However, the possibility remains that increased thiol abundance could be just enough to block cell death under certain circumstances. One way to demonstrate that this is not the case would be to overexpress or knockdown SLC7A11 in a GPx4 knock-out background, cultured with an antioxidant. If there is no change in the rate of cell death (upon antioxidant withdrawal) then the authors' hypothesis stands.

This reviewer asked a very insightful question. Recent studies showed that FSP1 operates independent of GPX4 to inhibit ferroptosis by converting coenzyme Q10 (CoQ) to its reduced form CoQH₂, which then acts as a radical trapping antioxidant to detoxify lipid peroxides [1, 2], and that coenzyme A (CoA) is also capable of inhibiting ferroptosis independent of GPX4 [3, 4]. Given that cysteine is a precursor for CoA biosynthesis and that CoA is also utilized in CoQ biosynthesis [5], it is likely that these recently identified anti-ferroptosis pathways represent GPX4-independent mechanisms linking cyst(e)ine to ferroptosis suppression. Therefore, it is indeed expected that SLC7A11 overexpression would provide some protection against ferroptosis in GPX4 knockout cells.

We want to emphasize that these GPX4-independent anti-ferroptosis mechanisms are not in conflict with our model that cysteine regulation of GPX4 protein synthesis plays a role in suppressing ferroptosis, because we are not proposing that this is the *only* mechanism (nor cysteine regulation of GSH and GPX4 levels are the only two mechanisms) linking cysteine to ferroptosis regulation. Indeed, in the Discussion of our manuscript, we have appropriately acknowledged and discussed these additional potential mechanisms. Importantly, in our study we provided key data to support our model that SLC7A11 regulation of GPX4 protein synthesis is at least one mechanism underlying how cysteine suppresses ferroptosis; that is, restoring GPX4 levels in SLC7A11 KO cells (to the level similar to that in control cells) partially restored cell viability under RSL3 treatment compared with that in control cells (Fig. 2j-l). As such, we feel that the experimental outcome from the suggested experiment will not provide additional support for nor counter-argument against our model. We modified our discussion on this part to better

reflect our point (see the second paragraph in page 13), and hope this reviewer will agree with us on this.

4. The authors find that BSO, unlike erastin or cystine deprivation, cannot decrease GPx4 expression. However, based on their later experiments, BSO should be more potent as a ferroptosis inducer with mTOR inhibition. Is this the case? Also, control experiments such as how BSO affects mTORC1 activity should be included.

5. While the lack of class II FINs for animal studies is a problem, using IKE is even more problematic in the context of this manuscript. Combining AZD8055, a potent antitumor drug in animals, with IKE, a potent SLC7A11 inhibitor, makes the PDX experiment unconvincing in terms of advancing the authors' argument (as IKE alone, even in the absence of AZD8055, should both deplete GSH and inhibit mTORC1, according to the mechanism proposed by the authors), although the combination therapy does show better effect than either single treatment. To directly support the hypothesis that mTOR inhibition can sensitize tumors to GPX4 inactivation, the authors have one good option: conduct the PDX experiment using BSO instead of IKE, +/- AZD. That is why the previous point is so critical.

Since questions 5 and 6 are both related to BSO, we would like to address them together. Our data showed that BSO treatment is not sufficient to induce obvious ferroptosis in the cell lines we have tested, which is consistent with other reports [6]. This reviewer reasoned that perhaps combining mTORC1 inhibitor (to decrease GPX4 level) and BSO (to deplete GSH) can further weaken the cellular defense system against ferroptosis and therefore induce more potent ferroptosis. We tested this interesting idea. As shown in **Rebuttal Figure 3a-b** (Figure S5g-h in our revised manuscript), combining Torin1 and BSO did not further induce lipid peroxidation or ferroptosis; as a control, we showed that RSL3 treatment potently induced lipid peroxidation and cell death. Furthermore, we showed that BSO treatment, unlike cystine starvation, did not reduce mTORC1 activation as gauged by S6K and S6 phosphorylation (**Rebuttal Figure 3c**, Figure S4b in our revised manuscript). As such, we did not find a clear rationale to further test Torin1 + BSO combo treatment in animal studies (as the reviewer suggested in question 6).

Rebuttal Figure 3. The effect of BSO on the regulation of lipid peroxidation, ferroptosis and mTOR1 signaling. a, UMR6 cells were treated with BSO (100 μ M), Torin1 (1 μ M) or BSO and Torin1 together for 24 hours or RSL3 (0.5 μ M) for 6 hours followed by lipid peroxidation analysis using Bodipy 581/591 C11. Bar graph showing quantitative analysis of lipid peroxidation levels. b, UMR6 cells were treated as described above followed by PI staining for cell death analysis. c, UMR6 cells were cultured in cystine-free or BSO-containing media for indicated time followed by Western blotting analysis. ns: not significant; **: P<0.01; ***: P<0.001; ****: P<0.0001.

This reviewer also asked a very insightful question that since IKE alone, even in the absence of AZD8055, already inhibits mTORC1, there seems to be little rationale to combine IKE and AZD8055 for animal studies. While these drugs can potently inhibit mTORC1 in cell culture studies, they only achieve moderate inhibition of the targets for in vivo studies (which is common for drug treatment in vivo). As shown in Fig. 4h in our manuscript, treatment with IKE or AZD8055 alone only had very moderate effect on reducing phosphor-4EBP1 or GPX4 level, but their combination resulted in a potent suppression of phosphor-4EBP1 and GPX4 levels. Therefore, for in vivo studies, AZD8055 can synergize with IKE to suppress GPX4 levels, thereby sensitizing tumors for IKE-induced ferroptosis. Consistent with this, we also observed a significantly increased 4HNE levels (a lipid peroxidation marker) in combo treatment compared with either treatment alone. We now added this description in page 12. With this interpretation, we hope this reviewer will agree that our therapeutic strategy indeed makes sense.

6. Does the withdrawal of other amino acids required for sustaining mTORC1 activity (such as leucine) also decrease GPx4 expression and enhance ferroptosis induced by class II FINs, as predicted by author's mechanism? Positive data will be a strong evidence supporting the proposed mechanism.

We thank the reviewer for asking this insightful question. As shown in **Rebuttal Figure 4a**, while removing cystine, serine, or leucine in culture medium all significantly suppressed mTORC1 activation, only cystine starvation, but not serine or leucine starvation, significantly decreased GPX4 protein levels. This is not entirely surprising: while diverse nutrient deprivation or metabolic stress suppresses mTORC1 signaling, it is expected that these mTORC1-suppressing conditions would not cause the exact same alterations in protein synthesis profiles, likely because their impact on mTORC1 signaling is quantitatively different, resulting in differential regulation of downstream targets in protein synthesis.

Rebuttal Figure 4. GPX4 and ferroptosis regulation by cystine, serine, and leucine. a, UMR6 cells were cultured in media without cystine, serine, or leucine for 24 hours followed by Western blotting analysis. b, UMR6 cells cultured in media without cystine, serine, or leucine were treated with indicated drugs (400 nM RSL3 or ML162, 5 μ M Ferrostatin-1) for 12 hours followed by cell viability analysis. **: $P < 0.05$; *: $P < 0.01$; ***: $P < 0.001$; ****: $P < 0.0001$.

As this reviewer suggested, we also tested the effect of other amino acid deprivation on ferroptosis. **Rebuttal Figure 4b** shows that serine or leucine deprivation also sensitized cells to RSL3- or ML162-induced ferroptosis, although the effect is somewhat more moderate than that of cystine starvation. Emerging data show that there is an intimate link between cellular metabolism and ferroptosis. Previous studies showed that glutamine or glucose deprivation

modulates ferroptosis sensitivity through regulating mitochondrial function or AMPK signaling [7-9]. It is possible that serine or leucine can also regulate ferroptosis sensitivity through additional mechanisms; however, it appears that their function in ferroptosis cannot be explained by regulating GPX4 levels. Considering this, we did not include these data in our current manuscript.

7. The tissue microarray in Figure 3j-k is unconvincing without another control protein, since 4EBP1 is a general regulator of translation.

This reviewer asked a great question. Since mTORC1-4EBP1 plays a general role in controlling protein synthesis, one could argue that anyway we will observe a correlation between GPX4 and p-4EBP1 levels in tissue microarray analysis (TMA).

		ACSL4		Fisher's exact test
		Negative/weak	Moderate/strong	P value
P-4EBP1	Negative/weak	57	23	0.0979
	Moderate/strong	14	13	

Rebuttal Fig. 5 The expression analysis of ACSL4 and p-4EBP1 in tissue microarrays.

A control protein whose level is not regulated by cyst(e)ine or mTORC1 can help address this question. Fig.1b in our manuscript shows that cystine starvation decreased GPX4 but not ACSL4 protein level. Therefore, we chose ACSL4 as the control protein to examine whether there is also a correlation between ACSL4 and p-4EBP1. As shown in **Rebuttal Figure 5** (Fig. S41 in our revised manuscript), there is no significant correlation between ACSL4 and p-4EBP1 levels from the same TMA, suggesting that the correlation between GPX4 and p-4EBP1 is relatively specific.

Minor points:

- In Figure 3A, the authors show the effect of cystine deprivation on mTOR targets. They are missing a panel for p-4EBP1; although the phosphorylation levels can reasonably be assumed from the size shift in 4EBP1, it may not be immediately clear to those unfamiliar with the protein, especially since p-4EBP1 is in the adjacent panels.

As this reviewer kindly suggested, we now added a p-4EBP1 blot to Figure 3A in our revised manuscript.

- In Figure 4A (left) there appear to be gates on the flow cytometry charts that do not correspond to the percentages shown.

We thank the reviewer for pointing this out. We corrected this in our revised manuscript.

Reviewer #2 (Remarks to the Author):

In this manuscript, Zhang et al. describes an unexpected role of cyst(e)ine mediated mTORC1 activation in regulating GPX4 protein synthesis. The authors showed that cyst(e)ine depletion inhibits the engagement of GPX4 mRNA with polysomes, and nicely ruled out the possibility

that Cys or mTORC1 is regulating GPX4 protein degradation. The alterations in GPX4 protein levels have direct consequences in the cellular sensitivity to ferroptosis, and authors revealed that co-inhibition of mTORC1 and GPX4 has synergistic effects in ferroptosis induction.

I am reading this paper with great interest, and I found the findings described here are novel and significant in the following ways:

1. Although the central role of GPX4 in safe-guarding cells from ferroptosis is well-appreciated, the regulation of GPX4 protein levels is poorly understood and largely overlooked. Previous reports in Zou et al., Nature Communications 2019 suggested that the GPX4 transcript levels are largely constant across more than 900 cancer cell lines from various lineages. This work is consistent with Zou et al. in showing that GPX4 mRNA levels are not changed by the chemical and genetic perturbations introduced here, and further describes the important findings on how cyst(e)ine availability is rate-limiting in GPX4 protein synthesis.

2. The exact mechanism of erastin-induced ferroptosis has not been fully resolved. This work suggests that in addition to blocking glutathione synthesis (thus reducing the abundances of the GPX4 co-factor), cyst(e)ine-depletion also inhibits GPX4 protein synthesis, suggesting a novel mechanism underlying the activity of erastin.

3. In this referee's lab, we have anecdotally noticed that mTORC1 inhibition sensitizes several cancer cell lines to GPX4 inhibition-induced ferroptosis, as well as the decreased GPX4 protein levels in response to erastin treatment in cancer cells. We were intrigued by these observations but did not understand the molecular mechanism underlying this connection. In this referee's opinion, the current study provides a very convincing explanation to the crosstalk between mTORC1 signaling and GPX4 activity.

4. The combinatorial strategy of ferroptosis induction coupled with mTORC1 inhibition is very appealing from a therapeutic perspective for augmenting the efficacy of ferroptosis-inducing approaches.

Overall, this study is performed with high rigor and the conclusions are largely supported by well-controlled experimental data. The manuscript is well-written and easy to understand. The conclusions in this study is expected have direct impact in the fields of ferroptosis, amino acid metabolism and mTORC1 signaling. It is appreciated that the authors validated each of their observations in multiple cell line models, and used cDNAs to rescue most of the genetic perturbations introduced in this study. Despite that how Cys regulates mTORC1 activity remains an intriguing scientific gap, this manuscript is complete on its own. This referee does not have additional experimental requests other than the minor suggestions listed below:

We thank this reviewer for positive comments and very insightful analyses of our manuscript.

1. The authors may briefly discuss (in the Discussion) their work in light of the recent reports in [Liu et al., Cancer Gene Therapy 2020 Interplay between MTOR and GPX4 signaling modulates autophagy dependent ferroptotic cancer cell death], and [Reinke et al., Plos ONE, 2014

Translational Regulation of GPx-1 and GPx-4 by the mTOR Pathway].

We thank this reviewer for the suggestion. A recent study showed rapamycin treatment can decrease GPX4 protein levels [10], whereas in our study rapamycin treatment did not obviously affect GPX4 protein levels. We noticed that the rapamycin concentration used in this study (25 μ M) was much higher than that used in our study (as well as in most other studies; typically within nM ranges). It is possible that rapamycin at such high concentrations can potentially inhibit 4E-BP1 phosphorylation and thereby suppress GPX4 protein synthesis. Another previous study indicated that mTORC1 regulates GPX4 protein translation [11]. However, this study showed that rapamycin treatment even increased GPX4 levels in the context of imatinib treatment. It remains unclear how mTORC1, a positive translation regulator, would suppress GPX4 protein synthesis in this context. Further studies are needed to clarify these questions. We now discussed and cited these relevant publications (see the second paragraph in page 14 in our manuscript).

Minor comments:

Figure 1a, the authors should specify whether multi-testing correction is performed with the p values presented in the volcano plot. If raw p value was presented here, this referee recommends using multiple-testing corrected data.

We thank reviewer for the suggestion to improve our statistical analysis. Accordingly, we have modified our analysis as described below. To identify significantly modulated proteins across starvation and control, we performed a Student t-Test with a permutation-based FDR cutoff of 0.05 and $S_0 = 0.1$. According to the new method of analysis, we now replaced Figure 1a with the updated one as shown in **Rebuttal Figure 6**. The corresponding method for analysis has been added to Mass spectrometry analysis in Method of our revised manuscript.

Rebuttal Figure 6. Mass spectrometry analysis to identify proteins differentially regulated upon cystine starvation.

Figure 1n, the label of the y-axis has overlapped text.

The label has been corrected in our revised manuscript.

Figure 2e, scale bar of the images is missing.

The scale bars have been added to all images in Figure 2e of our revised manuscript.

Figure 4a, there are several typos with regard to Torin1

These typos have been corrected in our revised manuscript.

Figure S4b, scale bar labeling, 100 μ M should be 100 μ m.

The labeling for this scale bar have been corrected in our revised manuscript (now Fig. S4c).

Line 228, typo in “Torrin1”

The typo has been corrected in our revised manuscript.

Reviewer #3 (Remarks to the Author):

In this manuscript, authors investigated mechanisms that regulate GPX4, an inhibitor of ferroptosis. The results show that cystine activates mTORC1 and promotes cap-dependent translation of GPX4 mRNA. Pharmacologic inhibition of mTORC1 decreases GPX4 protein synthesis and sensitizes cancer cells to ferroptosis. Overall, the studies are interesting that provide a possible mechanistic link between nutrient signaling and ferroptosis. However, some key findings are still preliminary that need to be strengthened before the manuscript is suitable for publication.

We appreciate the positive and insightful comments from this reviewer. We hope that our revision now has addressed the critiques from this reviewer.

1. Authors concluded that cystine activates mTORC1 through a Rag-dependent mechanism. However, the data are largely correlative. Authors should demonstrate that cystine regulates interaction of Rag with mTORC1 and that activation of mTORC1 by cystine is Rag-dependent.

This reviewer asked an important question. As shown in **Rebuttal Figure 7a-b** (Fig. 3e-f in our revised manuscript), RagA/B double knockout (DKO) in UMRC6 cells did not affect mTORC1 activation under normal culture conditions, which is consistent with previous observations made in RagA/B DKO MEFs [12]. Likewise, RagA/B deletion did not affect SLC7A11 or GPX4 level under normal culture

conditions. sgControl (sgC) and RagA/B DKO cells were then stimulated with cystine after they were deprived of cystine for 24 hours. Notably, RagA/B deficiency largely abolished cystine stimulation-induced mTORC1 activation and GPX4 expression.

Therefore, our data support a critical role of Rag in mediating cystine regulation of mTORC1 activation and GPX4 levels.

Rebuttal Figure 7. The role of Rag proteins in GPX4 protein regulation by cystine. a, Protein expression in UMRC6 control (sgC) and RagA/B double-knockout cells (DKO1 and DKO2) cells was analyzed by Western blotting. b, UMRC6 sgC and RagA/B DKO cells were cultured in media without cystine for 24 hours followed by stimulation with 200 μ M cystine and Western blotting analysis. c, UMRC6 cell cultured in media with or without cystine for 24 hours followed by pull-down using anti-raptor or normal IgG antibody. Then Raptor and RagB protein levels were analyzed by Western blotting.

We have also attempted to examine whether cystine stimulation can regulate Raptor-Rag interaction. However, despite multiple attempts, we have failed to detect the endogenous interaction between Raptor and Rag in UMRC6 cells (see **Rebuttal Figure 7c** for an example). It is possible that the interaction between Raptor and Rag is transient and weak in the cell lines used in our study (whereas previous studies have primarily used HEK293T cells to demonstrate amino acid-stimulated interaction between Raptor and Rag [13]). Since our data have convincingly established a critical role of Rag in cystine-stimulated mTORC1 activation and GPX4 regulation, we hope this reviewer will agree that the data demonstrating cystine-dependent Rag interaction with mTORC1 is not required for this manuscript.

2. Authors showed that treatment with Torin1 and AZD8055, not rapamycin decreases GPX4 protein level. Since Torin1 and AZD8055 are mTOR kinase inhibitors that target both mTORC1 and mTORC2, this observation suggests that mTORC2 is involved. Authors should address if mTORC2 plays a role in GPX4 regulation.

We thank reviewer for asking this important question. To examine the potential role of mTORC2 in regulation GPX4 levels, we deleted Rictor (a critical component of mTORC2) in UMRC6 cells. As shown in **Rebuttal Figure 8** (Figure S4i in revised manuscript), Rictor deletion significantly reduced phosphorylation levels of AKT as expected, but did not affect GPX4 levels. Our data therefore rule out the involvement of mTORC2 in regulating GPX4 levels.

3. It was previously shown that cystine regulates mTOR signaling through eIF2 α (Sci Rep 6: 30033). Authors should clarify whether eIF2 α is involved and how it is related to the proposed mechanism in the present study.

eIF2 α also plays an important role in coordinating amino acid availability with protein translational control through an amino acid-induced dephosphorylation of eIF2 α .

Consistent with this, we observed that cystine starvation significantly increased eIF2 α phosphorylation in UMRC6 cells (**Rebuttal Figure 9a**; Figure S4m in revised manuscript). It should be noted that, unlike mTOR phosphorylation (which activates mTOR), eIF2 α phosphorylation is inhibitory (therefore, cystine starvation decreases mTOR phosphorylation but increases eIF2 α phosphorylation). To address the potential role of eIF2 α phosphorylation in regulating GPX4 levels, we examined whether

inducing eIF2 α phosphorylation by treatment with salubrinal (an inhibitor of eIF2 α phosphatase) in the presence of cystine would mimic cystine starvation to decrease GPX4 levels. As shown in **Rebuttal Figure 9b** (Figure S4n in revised manuscript), while both cystine starvation and

Rebuttal Figure 8. Rictor deletion does not affect GPX4 protein levels. Protein levels in UMRC6 control (sgC) and Rictor knockout cells (sgRictor-1 and sgRictor-2) were analyzed by Western blotting.

Rebuttal Figure 9. The role of eIF2 α phosphorylation in GPX4 protein regulation. a, UMRC6 cells were cultured in media with indicated concentrations of cystine for 24 hours followed by Western blotting analysis. b, UMRC6 cell were cultured in cystine-free media or treated with 10 μ M salubrinal for 24 hours followed by Western blotting analysis.

salubrinal treatment increased eIF2 α phosphorylation, cystine starvation, but not salubrinal treatment, significantly decreased GPX4 levels. These data therefore suggest that cystine regulates GPX4 protein levels likely through eIF2 α -independent mechanisms, which is in line with our model that cystine regulates mTORC1 activation and GPX4 protein levels through Rag (see **Rebuttal Figure 7a-b**).

Reference

1. Bersuker, K., et al., *The CoQ oxidoreductase FSP1 acts parallel to GPX4 to inhibit ferroptosis*. Nature, 2019. **575**(7784): p. 688-692.
2. Doll, S., et al., *FSP1 is a glutathione-independent ferroptosis suppressor*. Nature, 2019. **575**(7784): p. 693-698.
3. Leu, J.I., M.E. Murphy, and D.L. George, *Mechanistic basis for impaired ferroptosis in cells expressing the African-centric S47 variant of p53*. Proc Natl Acad Sci U S A, 2019. **116**(17): p. 8390-8396.
4. Badgley, M.A., et al., *Cysteine depletion induces pancreatic tumor ferroptosis in mice*. Science, 2020. **368**(6486): p. 85-89.
5. Turunen, M., J. Olsson, and G. Dallner, *Metabolism and function of coenzyme Q*. Biochim Biophys Acta, 2004. **1660**(1-2): p. 171-99.
6. Lien, E.C., et al., *Glutathione biosynthesis is a metabolic vulnerability in PI(3)K/Akt-driven breast cancer*. Nat Cell Biol, 2016. **18**(5): p. 572-8.
7. Gao, M., et al., *Glutaminolysis and Transferrin Regulate Ferroptosis*. Mol Cell, 2015. **59**(2): p. 298-308.
8. Gao, M., et al., *Role of Mitochondria in Ferroptosis*. Mol Cell, 2019. **73**(2): p. 354-363 e3.
9. Lee, H., et al., *Energy-stress-mediated AMPK activation inhibits ferroptosis*. Nat Cell Biol, 2020. **22**(2): p. 225-234.
10. Liu, Y., et al., *Interplay between MTOR and GPX4 signaling modulates autophagy-dependent ferroptotic cancer cell death*. Cancer Gene Ther, 2020.
11. Reinke, E.N., et al., *Translational regulation of GPx-1 and GPx-4 by the mTOR pathway*. PLoS One, 2014. **9**(4): p. e93472.
12. Jewell, J.L., et al., *Metabolism. Differential regulation of mTORC1 by leucine and glutamine*. Science, 2015. **347**(6218): p. 194-8.
13. Sancak, Y., et al., *The Rag GTPases bind raptor and mediate amino acid signaling to mTORC1*. Science, 2008. **320**(5882): p. 1496-501.

REVIEWER COMMENTS

Reviewer #1 (Remarks to the Author):

The authors have done an excellent job in rebutting the reviewer critiques. In this reviewer's opinion, the manuscript is almost ready for publication, but there are a few questions related to the original comments #4, 5, 6 that need to be further addressed/clarified:

#4: BSO has been shown to be not as effective as erastin or cystine starvation for ferroptosis induction, and this manuscript might have provided a possible explanation: unlike the other two inducers that can both deplete glutathione and reduce GPx4 expression through inhibiting mTOR, BSO can only deplete glutathione. However, BSO plus inhibition of mTOR did not lead to potent ferroptosis induction, as the authors showed in revision. Because of this, a brief discussion on why BSO plus Torin is not effective and what might be other explanations will be helpful.

#5: In the animal experiment, the authors used a sub-optimal dose of AZD (did not effectively block 4EBP1 phosphorylation). While this condition works for the in vivo validation of the novel mechanism the authors uncovered, it is nonetheless not therapeutic relevant (why use such a non-effective dose of AZD? This won't happen in real clinical practice). The authors should at least why they used a low dose of AZD in the manuscript.

#6: The new result that the depletion of several other amino acids also led to complete inhibition of mTOR but had no effect on GPx4 expression is most concerning, as it directly challenges or contradicts the major conclusion of the manuscript: mTOR activity is required for GPx4 protein synthesis. This is a point that the authors should thoroughly address.

Reviewer #3 (Remarks to the Author):

Authors have largely addressed my concerns. I only have a minor comment that need to be incorporated into the final version.

Rebuttal Figure 7a: even though authors cited a paper that RagA/B double knockout did not diminish mTORC1 signaling (p-S6 and p-4EBP1) in MEFs, it still contradicts current paradigm and most of the high profile publications in the field. Moreover, in Figure 7b, there is clearly some stimulation of p-S6 by cystine in the absence of RagA/B. Authors should discuss on possible alternative mechanism(s) to RagA/B, in the main text and discussion sections. Cite literature support as appropriate.

Reviewer #4 (Remarks to the Author):

The manuscript suggests that a hitherto unrecognized regulatory mechanism to coordinate GPX4 protein synthesis with cyst(e)ine availability and suggesting use of mTORC1 inhibitors and FINs in cancer treatment. This is an important area of significant scientific impact and important clinical potential.

Major conclusions in this paper are based on documenting proteomic alterations associated with cystine starvation using a label-free quantification. Many of the conclusions depend on using mass spectrometry methods to document the changes in GPX4 protein synthesis. This review will focus on that key aspect of the methods and results.

The methods section of the paper goes to considerable minute details on many aspects of the methods and expands on many items that are relatively standard, e.g. that cell cultures contained penicillin (100 units/mL), streptomycin (100 µg/mL), or that extracted proteins were reduced with 5 mM dithiothreitol, alkylated with 15 mM iodoacetamide, and then quenched by 15 mM dithiothreitol (p. 16). We are told that samples were sequentially digested by Lys-C (enzyme: proteins, 1:100) for 4

hours and then trypsin (enzyme: proteins, 1:50) overnight at 37 °C pp. 16-17). Beyond these minute details of laboratory manipulations, there are, sadly, many broader explanations of the experimental design and the presentation of the results that are lacking.

Nowhere is it revealed how many biological replicates are included in the proteomics part of this experiment. The proteomic data are contained in Table S2 in a file called "256650_1_data_set_4982829_qhr4c4.xlsx." In that file columns A, B and C are described as Control_LFQ intensity and D, E and F as Starvation_LFQ intensity. These values are in a narrow range from 24 to 39. These data were generated by the MaxQuant program. In the reviewer's experience raw LFQ values are in the thousand to millions of counts, not 24 -39. Perhaps these are log transformed. We are left to wonder if they are transformed and whether it is base 2, base e, or base 10. This should be explicitly stated.

There is no explanation of how the Log2 Ratio Starvation_Control was calculated. This is made more ambiguous by of the uncertainties described in the next paragraph regarding biological replicates and p-values.

The text indicates at line 42 that "6 fractions were pooled from collected eluents using a previously reported method". There is no reference given. Reported to whom, when and how? It is unknown why multiple fractions were generated then pooled back together. What is the sense of fractionating and pooling back together? We have no idea how many biological replicates there were in this experiment. How can anyone evaluate the validity of the results and conclusions without even knowing how many replicates were used. We see only three control and three starvation columns in Table S2, but dozens of raw LC/MS files in the MassIVE repository upload cited (line 570). How these relate to 6 fractions and the unknown number of biological replicates is not stated.

In the previous rebuttal file provided to the reviewer (256650_1_rebuttal_4982823_qhrm6s_convrt.pdf). a reviewer asked that a multi-testing correction be applied to P-values. The authors responded by saying "we performed a Student t-Test with a permutation-based FDR cutoff of 0.05 and $S_0 = 0.1$ " This is reiterated in line 440 of the manuscript. This reviewer performed a T-test on the data in columns A-F in Table S2 (attached file "256650_1_data_set_4982829_qhr4c4_Review.xlsx" This T-test was the standard "garden variety" T-test included in Microsoft Excel with 2 tails, type 2 (equal variances/ homoscedastic). This test is reported in new column AE and the new -log in column AF (all bold blue). All 4,637 values in column AL are identical to six significant digits to the authors' column V which is titled "-Log Student's T-test p-value Starvation_Control" Thus the claimed multiple testing correction has not been applied. Even in Fig. 1a there is no indication that multiple testing correction has been applied as the values highlighted. If the correction has not been applied, most of the P-values reported may not be significant in this dataset with containing 4,637 tests.

It may be that the data are so weak, that if exposed in the bright light of the day, the claimed "unrecognized regulatory mechanism to coordinate GPX4 protein synthesis" is not supported by the results. The conclusion of this review is that the paper is not acceptable for publication.

Reviewer #1 (Remarks to the Author):

The authors have done an excellent job in rebutting the reviewer critiques. In this reviewer's opinion, the manuscript is almost ready for publication, but there are a few questions related to the original comments #4, 5, 6 that need to be further addressed/clarified:

We thank the reviewer for the positive comments on our manuscript. We hope that our revision now has addressed the remaining minor concerns from this reviewer.

#4: BSO has been shown to be not as effective as erastin or cystine starvation for ferroptosis induction, and this manuscript might have provided a possible explanation: unlike the other two inducers that can both deplete glutathione and reduce GPx4 expression through inhibiting mTOR, BSO can only deplete glutathione. However, BSO plus inhibition of mTOR did not lead to potent ferroptosis induction, as the authors showed in revision. Because of this, a brief discussion on why BSO plus Torin is not effective and what might be other explanations will be helpful.

We believe we have indeed discussed this point in the previous version of our manuscript (see below, which is copied from the discussion in page 13-14), and we apologize if we had not made this point more clearly.

“It is proposed that cyst(e)ine provides the rate-limiting precursor for the biosynthesis of GSH, which is subsequently used as the cofactor for GPX4-mediated lipid peroxidation detoxification and ferroptosis suppression [1]. However, in this study, we noted two major differences between cystine starvation and GSH depletion: (i) cystine starvation generally is much more potent than GSH depletion in inducing ferroptosis; (ii) cystine starvation, but not GSH depletion, sensitizes cells to class 2 FIN-induced ferroptosis. We propose that these differential effects can be explained by additional GSH-independent mechanisms downstream of cyst(e)ine in ferroptosis regulation, one of which is cyst(e)ine regulation of GPX4 protein synthesis as revealed in our current study. We want to emphasize that there exist additional ferroptosis defense mechanisms that operate in parallel to or independent of GPX4. For example, recent studies showed that FSP1 operates in parallel to GPX4 to inhibit ferroptosis by supplying coenzyme Q₁₀ (CoQ) [2, 3] and that coenzyme A (CoA) is also capable of inhibiting ferroptosis independent of GPX4 [4]. Given that cysteine is a precursor for CoA biosynthesis and that CoA is also utilized in CoQ biosynthesis [5], these recently identified anti-ferroptosis pathways represent additional mechanisms linking cyst(e)ine to ferroptosis regulation independent of GPX4.”

As discussed above, we propose that these other anti-ferroptosis mechanisms that operate downstream of cyst(e)ine but independent of GPX4, such as CoA and CoQ, can potentially explain why BSO plus Torin is not effective in inducing potent ferroptosis. To make this point more clearly, after the sentence “these recently identified anti-ferroptosis pathways represent additional mechanisms linking cyst(e)ine to ferroptosis regulation independent of GPX4”, we now add another statement “which likely explains the lack of strong ferroptosis induction in cells treated with mTORC1 inhibitor and BSO (to suppress both GPX4 and GSH synthesis).”

#5: In the animal experiment, the authors used a sub-optimal dose of AZD (did not effectively

block 4EBP1 phosphorylation). While this condition works for the in vivo validation of the novel mechanism the authors uncovered, it is nonetheless not therapeutic relevant (why use such a non-effective dose of AZD? This won't happy in real clinical practice). The authors should at least why they used a low dose of AZD in the manuscript.

We would like to clarify that we used the same dose of AZD8055 (10 mg/kg) in our animal studies as used in the original publication on AZD8055 [6] and multiple other studies (for example, see [7-11]); therefore, we were not deliberately choosing a sub-optimal dose of AZD8055 in our animal studies. Further increasing AZD8055 doses likely will cause unwanted toxicity issues in animals. Because of tumor heterogeneity, it is quite common that one dose of drug is effective in inhibiting its target in one tumor model yet becomes non-effective in another model (or moderately effective, as shown in our preclinical model). This indeed represents one significant challenge in cancer therapies, and has further motivated combination therapies to better target mTORC1 (as shown in our current study). To follow this reviewer's kind suggestion, we now incorporated these points into our manuscript (see page 16).

#6: The new result that the depletion of several other amino acids also led to complete inhibition of mTOR but had no effect on GPx4 expression is most concerning, as it directly challenges or contradicts the major conclusion of the manuscript: mTOR activity is required for GPx4 protein synthesis. This is a point that the authors should thoroughly address.

We thank the reviewer for asking this insightful question. We acknowledge that we currently do not understand this regulation selectivity on the mechanistic level, but want to point out that, at the conceptual level, this is quite common in biological systems. As illustrated in **rebuttal letter Fig. 1**, mTORC1 receives signaling inputs from diverse stimuli, including amino acids (such as cystine and leucine), glucose, and growth factors. Once activated, mTORC1 functions to promote protein synthesis of genes GPX4, a, b, c, and so on. Removal of cystine, leucine, glucose, or growth factors (as well as mTORC1 inhibitor treatment) can all potently suppress mTORC1 activity (as gauged by S6K/4EBP1 phosphorylation); however, these nutrient deprivation conditions would not cause the suppression of protein synthesis on the EXACT same list of target genes. For example, it has been shown that starvation of different amino acids results in differential ribosome occupancies on specific genes [12]. Several mechanisms can account for such a selectivity in signaling regulation:

(i) These upstream stimuli can differentially affect the levels of downstream targets through additional mechanisms other than mTORC1. For example, it is possible that, while both leucine and cystine starvation decrease GPX4 protein synthesis by inactivating mTORC1, leucine starvation (but not cystine starvation) might also induce adaptive responses to restore GPX4

Fig. 1. A schematic depicting how mTORC1 receives diverse upstream stimuli to control the protein synthesis of different downstream target genes.

protein level, resulting in differential GPX4 protein level change upon cystine or leucine starvation.

(ii) Scaffold proteins can limit specific upstream regulators to downstream effectors. This is well established in MAPK signaling. Notably, a recent study showed that amino acid starvation, but not serum starvation, selectively modulates mTORC1 regulation of downstream effector TFEB through similar mechanisms [13]. Whether such mechanisms are also involved in cystine regulation of GPX4 protein synthesis remains to be examined.

(iii) These upstream stimuli can regulate mTORC1 signaling quantitatively differently; however, such quantitative differences might not be readily revealed by biochemical readouts such as S6K/4EBP1 phosphorylation. This could simply reflect our current knowledge or technical limitations. As another example, whereas diverse upstream stimuli, such as cell-cell contact, serum starvation, and energy stress, can all potentially inactivate YAP (as gauged by YAP phosphorylation), these stimuli do not regulate the EXACT same list of YAP1 target genes. Instead, some core YAP target genes are similarly regulated, whereas many other target genes are differentially impacted, by different upstream stimuli of the Hippo-YAP pathway, although the underlying mechanisms still remain poorly understood [14].

As our current study is not set out to understand the differential effects of cystine and other amino acids on mTORC1 signaling, we hope this reviewer will agree that addressing these questions is beyond the scope of this study. We have incorporated these discussions into our manuscript (see the second paragraph on page 15, and Fig. S7), and hope that this will stimulate future studies on this interesting question.

Reviewer #3 (Remarks to the Author):

Authors have largely addressed my concerns. I only have a minor comment that need to be incorporated into the final version.

We thank the reviewer for the positive comments on our manuscript. We hope that our revision now has addressed the remaining minor concern from this reviewer.

Rebuttal Figure 7a: even though authors cited a paper that RagA/B double knockout did not diminish mTORC1 signaling (p-S6 and p-4EBP1) in MEFs, it still contradicts current paradigm and most of the high profile publications in the field. Moreover, in Figure 7b, there is clearly some stimulation of p-S6 by cystine in the absence of RagA/B. Authors should discuss on possible alternative mechanism(s) to RagA/B, in the main text and discussion sections. Cite literature support as appropriate.

We thank the reviewer for asking this insightful question. Previous studies showed that acute deletion of RagA in MEFs significantly decreased basal mTORC1 signaling [15] but MEFs with stable RagA/B deletion only exhibited very moderate reduction in mTORC1 signaling under normal culture conditions [16]. Another recent study showed that knocking down Rag A or C did not affect basal mTORC1 activation in Hela cells [17] (see Fig. 1C and D in this study). RagA/B

deletion in heart even slightly increased mTORC1 signaling [18]. Previous studies noted that Rag A is essential for cell growth in MEFs [15]. Likewise, we also noticed that it is challenging to generate RagA/B complete KO UMRC6 cells from single clones using CRISPR approaches. In our study, we have used pooled CRISPR RagA/B KO cells, which maintain residual Rag A and B expression (see **Rebuttal Letter Fig. 2**; Fig. 3e in our manuscript). It is possible that residual RagA/B expression is required to maintain mTORC1 activation and cell growth under basal conditions, and/or RagA/B deficiency induces compensatory mTORC1 activation in order to allow for long-term culture of these RagA/B KO cells, explaining the lack of obvious mTORC1 signaling reduction in our RagA/B KO cells under normal culture conditions. We reason that the residual Rag expression and/or the compensatory mTORC1 activation in Rag KO cells likely accounts for the discrepancies from various studies on the differential effects of acute vs stable deficiency of Rags on basal mTORC1 activation.

Apparently, this residual RagA/B expression is not sufficient to maintain cystine-induced mTORC1 activation, resulting in a significant attenuation of cystine stimulation-induced mTORC1 activation in our RagA/B KO cells. However, as this reviewer kindly pointed out, cystine-induced mTORC1 activation is not completely abolished in our RagA/B KO cells, and we acknowledge that there might exist additional mechanisms mediating cystine-induced mTORC1 activation, which is consistent with previous findings demonstrating Rag-independent but Arf-1- or Rab-1A-dependent mechanisms to mediate mTORC1 activation in response to some amino acids [16, 19]. It will be interesting to examine whether Arf-1 or Rab1A is also involved in regulating cystine-induced mTORC1 activation in future studies.

We have now added this discussion into our manuscript and cited relevant publications (see page 14-15).

Reviewer #4 (Remarks to the Author):

The manuscript suggests that a hitherto unrecognized regulatory mechanism to coordinate GPX4 protein synthesis with cyst(e)ine availability and suggesting use of mTORC1 inhibitors and FINs in cancer treatment. This is an important area of significant scientific impact and important clinical potential.

Major conclusions in this paper are based on documenting proteomic alterations associated with cystine starvation using a label-free quantification. Many of the conclusions depend on using mass spectrometry methods to document the changes in GPX4 protein synthesis. This review will focus on that key aspect of the methods and results.

Fig. 2. RagA/B double-knockout (DKO) UMRC6 cells still exhibit residual expression of RagA and RagB. LE: long exposure.

We thank the reviewer for the comments and analysis of the proteomic data presented in this manuscript (more specifically, Fig. 1a and S1a). We hope our responses below have now addressed all the critiques from this reviewer.

The methods section of the paper goes to considerable minute details on many aspects of the methods and expands on many items that are relatively standard, e.g. that cell cultures contained containing penicillin (100 units/mL), streptomycin (100 µg/mL), or that extracted proteins were reduced with 5 mM dithiothreitol, alkylated with 15 mM iodoacetamide, and then quenched by 15 mM dithiothreitol (p. 16). We are told that samples were sequentially digested by Lys-C (enzyme: proteins, 1:100) for 4 hours and then trypsin (enzyme: proteins, 1:50) overnight at 37 °C pp. 16-17). Beyond these minute details of laboratory manipulations, there are, sadly, many broader explanations of the experimental design and the presentation of the results that are lacking.

We agree with this reviewer that the Methods section should include detailed experimental conditions so that other researchers can replicate our findings in the future. Following the kind suggestion of this reviewer, we now provide a flowchart of our proteomic experimental design, which we hope can help the readers to understand broadly our experimental approach (**Rebuttal Figure 3**; Figure S1a in our revised manuscript). Our response below will provide additional clarifications and answers to this reviewer's questions.

Fig. 3. The flowchart of experimental design of our proteomic studies.

Nowhere is it revealed how many biological replicates are included in the proteomics part of this experiment. The proteomic data are contained in Table S2 in a file called “256650_1_data_set_4982829_qhr4c4.xlsx.” In that file columns A, B and C are described as Control_LFQ intensity and D, E and F as Starvation_LFQ intensity. These values are in a narrow range from 24 to 39. These data were generated by the MaxQuant program. In the reviewer's experience raw LFQ values are in the thousand to millions of counts, not 24 -39. Perhaps these are log transformed. We are left to wonder if they are transformed and whether it is base 2, base e, or base 10. This should be explicitly stated.

We thank the reviewer for pointing this out. In the previous version of this manuscript, we stated “samples in triplicates” in the Figure legend of Supplementary Fig. 1a (now Supplementary Fig. 1b). To further clarify the issue of biological replicates, we now stated this (cells in triplicate) in the Methods section (under “Mass spectrometry analysis”, page 18), and also provided it in the experimental flowchart above (Figure S1a in our revised manuscript).

The reviewer is correct pointing out that the values presented in Table S2 are log transformed values. As kindly suggested by this reviewer, we have now changed the titles of corresponding columns in Table S2 from “XXX_LFQ intensity XX” to “XXX_Log2 LFQ intensity XX”. As stated in Methods section (page 17), we used Perseus to conduct the proteomics data analysis, which was developed by Matthias Mann and Jurgen Cox [20]. This method is widely used in proteomics data analysis. When the Perseus was used for data analysis, we followed the tutorial in the official Perseus website and older version tutorial (http://coxdocs.org/doku.php?id=perseus:user:use_cases:start, http://Inbio.cnpem.br/wp-content/uploads/2012/11/Tutorial-Perseus_02062015_release_v1.pdf) to perform log2 transformation before further analysis. The transfer of original MaxQuant values to Log2 data for further analysis has been adopted by others [21]. Indeed, log2 LFQ intensity data were presented in other publications [22, 23].

There is no explanation of how the Log2 Ratio Starvation_Control was calculated. This is made more ambiguous by of the uncertainties described in the next paragraph regarding biological replicates and p-values.

As mentioned above, we have now stated the three biological replicates in the legend of Fig. S1b and also in Methods section. The Log2 Ratio was calculated using Perseus following the standard instruction [24] and the tutorial (http://coxdocs.org/doku.php?id=perseus:user:use_cases:start, http://Inbio.cnpem.br/wp-content/uploads/2012/11/Tutorial-Perseus_02062015_release_v1.pdf). Because Perseus is widely used in the proteomic data analysis by different groups [22, 23, 25-31] (with more than 2350 citations from Google Scholar) and we are following the standard procedure, we did not provide the detailed description on the calculation of fold change, but only showed the cutoff criteria for significantly changed proteins used in this study, which is also consistent with other publications as cited above. Following the comment of this reviewer, we have now cited Perseus instruction and tutorial in the revised Methods section to provide more details for the readers (reference 61 in the manuscript).

The text indicates at line 42 that “6 fractions were pooled from collected eluents using a previously reported method”. There is no reference given. Reported to whom, when and how? It is unknown why multiple fractions were generated then pooled back together. What is the sense of fractionating and pooling back together? We have no idea how many biological replicates there were in this experiment. How can anyone evaluate the validity of the results and conclusions without even knowing how many replicates were used. We see only three control and three starvation columns in Table S2, but dozens of raw LC/MS files in the MassIVE repository upload cited (line 570). How these relate to 6 fractions and the unknown number of biological replicates is not stated.

We apologize for not stating this more clearly in the previous version of our manuscript. In this study, we used three biological replicates, with 6 fractions being pooled for each biological replicate. Following the suggestion of this reviewer, we have now provided the original citation

on the HPLC fractionation and pooling of fractions [32] in the revised Methods section (reference 60 in the manuscript). It should be noted that fractionating the whole proteome/phosphoproteome (to reduce sample complexity and to detect low-abundance proteins) has now been adopted for both labeling and label-free proteomics analyses in proteomics field. This approach has been described in Clinical Proteomics Tumor Analysis Consortium (CPTAC) protocol and other publications [32-37]. Additionally, fractions are often pooled to reduce mass spectrometry running time, for example pool 96 fractions into 24 or 12 fractions. This strategy enables researchers to obtain more proteins or PTM IDs within the constrain of mass spectrometry time. The strategy is further illustrated in **Rebuttal Letter Fig. 4**, which was copied from Fig. 2 in [32].

Fig. 4. Illustration of sample preparation for MS analysis.

*In the previous rebuttal file provided to the reviewer (256650_1_rebuttal_4982823_qhrm6s_convrt.pdf), a reviewer asked that a multi-testing correction be applied to P-values. The authors responded by saying “we performed a Student t-Test with a permutation-based FDR cutoff of 0.05 and $S_0 = 0.1$ ” This is reiterated in line 440 of the manuscript. This reviewer performed a T-test on the data in columns A-F in Table S2 (attached file “256650_1_data_set_4982829_qhr4c4_Review.xlsx” This T-test was the standard “garden variety” T-test included in Microsoft Excel with 2 tails, type 2 (equal variances/homoscedastic). This test is reported in new column AE and the new $-\log$ in column AF (all bold blue). All 4,637 values in column AL are identical to six significant digits to the authors’ column V which is titled “-Log Student's T-test p-value Starvation_Control” **Thus the claimed multiple testing correction has not been applied.** Even in Fig. 1a there is no indication that multiple testing correction has been applied as the values highlighted. If the correction has not been applied, most of the P-values reported may not be significant in this dataset with containing 4,637 tests.*

We thank the reviewer for asking this question. As described in Methods section, we used Perseus for the analysis. Perseus is a well-established software for proteomics analysis. The $-\log$ p-value, Q-value and Student's T-test statistical analyses were performed via Perseus. Our analysis is similar to proteomic analyses reported in other recent publications [22, 23, 25, 26, 28, 30, 31]. In the previous version of our manuscript, we used p-value and fold change as criteria to decide significantly changed proteins. Following the suggestion of reviewer #2 for *multi-testing correction*, in the revised manuscript we generated q value for each p value regarding the analysis of identified proteins, as highlighted by red color in the Rebuttal Table 1 shown below (Supplementary Table S2 in our manuscript). A q -value is a p -value that has been adjusted for the False Discovery Rate (FDR). The False Discovery Rate approach to p -values assigns an

-Log Student's T-test p-value Starvation_Control	Student's T-test q-value Starvation_Control	Log2 Ratio Starvation_Control	Student's T	Protein IDs	Majority pr	Protein nar	Gene name id
4.234872195	0	-1.056161245	-6.63181	D6RB81;Q6 D6RB81;Q6	Alpha-met	AMACR	887
5.242946235	0	-0.794553121	-6.36339	Q8WXH0;C Q8WXH0;C	Nesprin-2	SYNE2	5053
3.73194889	0	1.184992472	6.263983	P13995;P1 P13995;P1	Bifunction:	MTHFD2	2436
4.187954743	0	1.483480453	7.995668	P09601;B1 P09601;B1	Heme oxyg	HMOX1	2330
4.20716442	0	2.289484024	9.930914	P08243-2;I P08243-2;I	Asparagin	ASNS	2301
2.504031402	0.006736842	1.579705556	4.534341	Q14061;H; Q14061;H;	Cytochrom	COX17	3748
3.153692628	0.007111111	-0.87600708	-4.54386	P46013;P4 P46013;P4	Antigen KI-	MKI67	2925
2.460119647	0.007529412	1.754427592	4.574533	O00458;O0 O00458;O0	Interferon-	IFRD1	1589
3.321617373	0.008	0.849787394	4.683253	Q15758;M Q15758;M	Neutral am	SLC1A5	3958
2.442096738	0.008533333	-2.089710871	-4.73285	Q12882;Q1 Q12882	Dihydropy	DPYD	3596
3.894878279	0.009142857	0.730517069	4.87101	P07602;C9 P07602;C9	Prosaposin	PSAP	2275
3.597372535	0.009846154	0.850730896	5.025373	Q9Y617;Q6 Q9Y617;Q6	Phosphose	PSAT1	6789
3.312322315	0.010666667	0.980314255	5.040616	Q99519;E9 Q99519;E9	Sialidase-1	NEU1	5487
3.402026405	0.011130435	-0.734525045	-4.39495	A0A0X1K6 A0A0X1K6	Cordon-ble	COBLL1	340
2.726323228	0.011636364	-1.122111638	-4.42064	K7ERP4;A0 K7ERP4;A0	Glutathion	GPX4	20
3.977751425	0.011636364	0.803720474	5.274392	P53634;H; P53634	Dipeptidyl	CTSC	3147
4.196150168	0.012190476	0.593900045	4.429427	P41250;H7 P41250	Glycine-tR	GARS	2879
2.706572444	0.0128	-1.165859222	-4.45366	Q96FL9-4;Q Q96FL9-4;Q	Polypeptid	GALNT14	5291
3.443775124	0.0128	1.186173757	5.764747	O43715 O43715	TP53-regul	TRIAP1	1799
4.018878583	0.014222222	0.925989787	5.826797	Q16822;Q1 Q16822;Q1	Phosphoen	CK2	4006
2.809675925	0.014769231	-0.935894648	-4.21672	Q8N9V3;Q1 Q8N9V3;Q1	WD repeat,	WDSUB1	4823
3.012848585	0.01536	-0.823280334	-4.22455	Q9UDY2-3; Q9UDY2-3;	Tight junct	TJP2	6391
5.065715383	0.016	0.510916392	4.340418	A0A286YF2 A0A286YF2	D-3-phosph	PHGDH	400
4.032012225	0.016	0.955507914	5.959508	P08195-2;I P08195-2	4F2 cell-su	SLC3A2	2298
3.216394291	0.018285714	-1.64172554	-6.13695	P26447 P26447	Protein S1c	S100A4	2659
2.854290303	0.018814815	0.835413615	4.056452	P23381;P2 P23381;P2	Tryptophan	WARS	2610
4.692473835	0.021333333	-0.839103699	-6.1654	Q13501;E7 Q13501;E7	Sequestoso	SQSTM1	3687
5.496327888	0.039428571	-0.44171524	-3.94595	A0A0D95F1 A0A0D95F1	Unconvent	MYO18A	285
3.062368743	0.057263158	-0.559478124	-3.44105	Q9NTJ3;E9 Q9NTJ3;E9	Structural i	SMMC4	6122
3.384585258	0.057333333	-0.572805405	-3.74645	P49736;HC P49736;HC	DNA replici	MCM2	3014
3.726993835	0.058580645	-0.498853048	-3.62407	P49327;A0 P49327;A0	Fatty acid s	FASN	2993
2.938957045	0.058810811	-0.596932729	-3.47207	A0A499F14 A0A499F14			545
2.17006615	0.058909091	1.169989268	3.57437	Q01650 Q01650	Large neutr	SLC7A5	3461
3.645536138	0.059310345	0.533063889	3.748619	P07686;Q5 P07686;Q5	Beta-hexos	HEXB	2278

Table 1. Represented statistics of mass spec data including p value and q value.

adjusted p -value for each test. q -value has been described in previous publication [38]. This q -value was calculated based on permutation-based FDR cutoff of 0.05 and $S_0 = 0.1$ by Perseus automatically. As stated in Methods section, we used this value, i.e. permutation-based FDR cutoff of 0.05 and $S_0 = 0.1$, but not p -value, as criteria to determine the significantly regulated proteins upon treatment. Permutation-based FDR is used to adjust for multiplicity tests by controlling the family-wise type I error rate (FWER) - which is the probability of making one or more false discoveries, or type I errors among all the hypotheses when performing multiple hypothesis tests-without assuming t distribution of the test statistics of each gene's differential expression. With the added S_0 , it is the Significance Analysis of Microarrays (SAM) test [39]. Our analysis was based on the instruction in official Perseus website and the tutorial (http://coxdocs.org/doku.php?id=perseus:user:use_cases:interactions, <http://coxdocs.org/doku.php?id=perseus:user:activities:matrixprocessing:tests:twosamplettestprocessing>, http://lnbio.cnpem.br/wp-content/uploads/2012/11/Tutorial-Perseus_02062015_release_v1.pdf), and is consistent with other analyses from recent publications [22, 23, 25, 26, 28, 30, 31], which also used permutation-based FDR at a given S_0 as the cutoff criteria.

We should point out that in our Fig.1a (shown as **Rebuttal Letter Fig. 5a**), $-\text{Log } p$ value was set as Y axis, but the cutoff criteria still are $FDR < 0.05$ and $S_0 = 0.1$ as described in our manuscript. This is due to the fact that the q -value of some proteins (e.g. top 4 targets

Fig. 5 Deregulated genes from Treatment vs Control shown in the plot generated by $-\text{Log } p$ value (a) or $-\text{Log } q$ value (b) for Y

in Table 1) was 0; if we use -Log q value as Y axis, these top proteins would not show up in the Y axis (see **Rebuttal Letter Fig. 5b**). It is acceptable to use -Log p value as Y axis in proteomics studies as cited here [22, 23, 25, 26, 28, 31].

It may be that the data are so weak, that if exposed in the bright light of the day, the claimed “unrecognized regulatory mechanism to coordinate GPX4 protein synthesis” is not supported by the results. The conclusion of this review is that the paper is not acceptable for publication.

In summary, we have applied standard analysis tools that are commonly used in proteomics field in this study. We hope our responses above have addressed the concerns from this reviewer.

References:

1. Stockwell, B.R., et al., *Ferroptosis: A Regulated Cell Death Nexus Linking Metabolism, Redox Biology, and Disease*. Cell, 2017. **171**(2): p. 273-285.
2. Bersuker, K., et al., *The CoQ oxidoreductase FSP1 acts parallel to GPX4 to inhibit ferroptosis*. Nature, 2019. **575**(7784): p. 688-692.
3. Doll, S., et al., *FSP1 is a glutathione-independent ferroptosis suppressor*. Nature, 2019. **575**(7784): p. 693-698.
4. Leu, J.I., M.E. Murphy, and D.L. George, *Mechanistic basis for impaired ferroptosis in cells expressing the African-centric S47 variant of p53*. Proc Natl Acad Sci U S A, 2019. **116**(17): p. 8390-8396.
5. Turunen, M., J. Olsson, and G. Dallner, *Metabolism and function of coenzyme Q*. Biochim Biophys Acta, 2004. **1660**(1-2): p. 171-99.
6. Chresta, C.M., et al., *AZD8055 is a potent, selective, and orally bioavailable ATP-competitive mammalian target of rapamycin kinase inhibitor with in vitro and in vivo antitumor activity*. Cancer Res, 2010. **70**(1): p. 288-98.
7. Rosborough, B.R., et al., *Adenosine triphosphate-competitive mTOR inhibitors: a new class of immunosuppressive agents that inhibit allograft rejection*. Am J Transplant, 2014. **14**(9): p. 2173-80.
8. Caumanns, J.J., et al., *Integrative Kinome Profiling Identifies mTORC1/2 Inhibition as Treatment Strategy in Ovarian Clear Cell Carcinoma*. Clin Cancer Res, 2018. **24**(16): p. 3928-3940.
9. Zhou, M., et al., *Boosting mTOR-dependent autophagy via upstream TLR4-MyD88-MAPK signalling and downstream NF-kappaB pathway quenches intestinal inflammation and oxidative stress injury*. EBioMedicine, 2018. **35**: p. 345-360.
10. Chang, L., et al., *A low dose of AZD8055 enhances radiosensitivity of nasopharyngeal carcinoma cells by activating autophagy and apoptosis*. Am J Cancer Res, 2019. **9**(9): p. 1922-1937.
11. Rosas-Plaza, X., et al., *Dual mTORC1/2 Inhibition Sensitizes Testicular Cancer Models to Cisplatin Treatment*. Mol Cancer Ther, 2020. **19**(2): p. 590-601.
12. Subramaniam, A.R., B.M. Zid, and E.K. O'Shea, *An integrated approach reveals regulatory controls on bacterial translation elongation*. Cell, 2014. **159**(5): p. 1200-1211.

13. Napolitano, G., et al., *A substrate-specific mTORC1 pathway underlies Birt-Hogg-Dube syndrome*. *Nature*, 2020. **585**(7826): p. 597-602.
14. Ma, S., et al., *The Hippo Pathway: Biology and Pathophysiology*. *Annu Rev Biochem*, 2019. **88**: p. 577-604.
15. Efeyan, A., et al., *RagA, but not RagB, is essential for embryonic development and adult mice*. *Dev Cell*, 2014. **29**(3): p. 321-9.
16. Jewell, J.L., et al., *Metabolism. Differential regulation of mTORC1 by leucine and glutamine*. *Science*, 2015. **347**(6218): p. 194-8.
17. Yang, S., et al., *The Rag GTPase Regulates the Dynamic Behavior of TSC Downstream of Both Amino Acid and Growth Factor Restriction*. *Dev Cell*, 2020. **55**(3): p. 272-288 e5.
18. Kim, Y.C., et al., *Rag GTPases are cardioprotective by regulating lysosomal function*. *Nat Commun*, 2014. **5**: p. 4241.
19. Thomas, J.D., et al., *Rab1A is an mTORC1 activator and a colorectal oncogene*. *Cancer Cell*, 2014. **26**(5): p. 754-69.
20. Tyanova, S., et al., *The Perseus computational platform for comprehensive analysis of (prote)omics data*. *Nat Methods*, 2016. **13**(9): p. 731-40.
21. Cox, J. and M. Mann, *ID and 2D annotation enrichment: a statistical method integrating quantitative proteomics with complementary high-throughput data*. *BMC Bioinformatics*, 2012. **13 Suppl 16**: p. S12.
22. Sacco, F., et al., *Phosphoproteomics Reveals the GSK3-PDX1 Axis as a Key Pathogenic Signaling Node in Diabetic Islets*. *Cell Metab*, 2019. **29**(6): p. 1422-1432 e3.
23. Deshmukh, A.S., et al., *Proteomics-Based Comparative Mapping of the Secretomes of Human Brown and White Adipocytes Reveals EPDR1 as a Novel Batokine*. *Cell Metab*, 2019. **30**(5): p. 963-975 e7.
24. Tyanova, S. and J. Cox, *Perseus: A Bioinformatics Platform for Integrative Analysis of Proteomics Data in Cancer Research*. *Methods Mol Biol*, 2018. **1711**: p. 133-148.
25. Tanzer, M.C., et al., *Quantitative and Dynamic Catalogs of Proteins Released during Apoptotic and Necroptotic Cell Death*. *Cell Rep*, 2020. **30**(4): p. 1260-1270 e5.
26. Sacco, F., et al., *Deep Proteomics of Breast Cancer Cells Reveals that Metformin Rewires Signaling Networks Away from a Pro-growth State*. *Cell Syst*, 2016. **2**(3): p. 159-71.
27. Bruning, F., et al., *Sleep-wake cycles drive daily dynamics of synaptic phosphorylation*. *Science*, 2019. **366**(6462).
28. Pozniak, Y., et al., *System-wide Clinical Proteomics of Breast Cancer Reveals Global Remodeling of Tissue Homeostasis*. *Cell Syst*, 2016. **2**(3): p. 172-84.
29. Oberhuber, M., et al., *STAT3-dependent analysis reveals PDK4 as independent predictor of recurrence in prostate cancer*. *Mol Syst Biol*, 2020. **16**(4): p. e9247.
30. Tyanova, S., et al., *Proteomic maps of breast cancer subtypes*. *Nat Commun*, 2016. **7**: p. 10259.
31. Murgia, M., et al., *Proteomics of Cytochrome c Oxidase-Negative versus -Positive Muscle Fiber Sections in Mitochondrial Myopathy*. *Cell Rep*, 2019. **29**(12): p. 3825-3834 e4.

32. Batth, T.S. and J.V. Olsen, *Offline High pH Reversed-Phase Peptide Fractionation for Deep Phosphoproteome Coverage*. Methods Mol Biol, 2016. **1355**: p. 179-92.
33. Mertins, P., et al., *Reproducible workflow for multiplexed deep-scale proteome and phosphoproteome analysis of tumor tissues by liquid chromatography-mass spectrometry*. Nat Protoc, 2018. **13**(7): p. 1632-1661.
34. Wang, Y., et al., *Reversed-phase chromatography with multiple fraction concatenation strategy for proteome profiling of human MCF10A cells*. Proteomics, 2011. **11**(10): p. 2019-26.
35. Mertins, P., et al., *Integrated proteomic analysis of post-translational modifications by serial enrichment*. Nat Methods, 2013. **10**(7): p. 634-7.
36. Batth, T.S., C. Francavilla, and J.V. Olsen, *Off-line high-pH reversed-phase fractionation for in-depth phosphoproteomics*. J Proteome Res, 2014. **13**(12): p. 6176-86.
37. Dwivedi, R.C., et al., *Practical implementation of 2D HPLC scheme with accurate peptide retention prediction in both dimensions for high-throughput bottom-up proteomics*. Anal Chem, 2008. **80**(18): p. 7036-42.
38. Storey, J.D. and R. Tibshirani, *Statistical significance for genomewide studies*. Proc Natl Acad Sci U S A, 2003. **100**(16): p. 9440-5.
39. Tusher, V.G., R. Tibshirani, and G. Chu, *Significance analysis of microarrays applied to the ionizing radiation response*. Proc Natl Acad Sci U S A, 2001. **98**(9): p. 5116-21.

REVIEWERS' COMMENTS

Reviewer #1 (Remarks to the Author):

The authors have addressed all my comments.

Reviewer #4 (Remarks to the Author):

The authors have responded to review comments and improved the manuscript text as well as Supplementary Table S2. This has effectively responded to the criticisms made and the paper should be published.

One new point generated is that the authors indicate in the rebuttal letter that in response to the criticism of the experimental design not well described in the original manuscript they have created a new 'Supplementary Figure 1...a The flowchart of experimental design for proteomic studies.' They do mention in the rebuttal letter that Fig. S1a is derived from reference 60 (Batth TS, Olsen JV. Offline High pH Reversed-Phase Peptide Fractionation for Deep Phosphoproteome Coverage. *Methods Mol Biol* 1355, 179-192 (2016).

However neither in the text of the manuscript nor in the figure description do they tell us that Fig. S1a is copied directly from Fig. 2 of Batth and Olsen (2016) (attached to this review). This would appear to violate Nature Communications policy as it does not provide appropriate and unambiguous attribution. Moreover, in this case since it is a verbatim copy of part of an image from another paper, permission to reproduce that image would typically need to be obtained from the copyright holder.

Detailed Point-by-point response to the reviewer's comments:

Reviewer #4 (Remarks to the Author):

The authors have responded to review comments and improved the manuscript text as well as Supplementary Table S2. This has effectively responded to the criticisms made and the paper should be published.

One new point generated is that the authors indicate in the rebuttal letter that in response to the criticism of the experimental design not well described in the original manuscript they have created a new 'Supplementary Figure 1...a The flowchart of experimental design for proteomic studies.' They do mention in the rebuttal letter that Fig. S1a is derived from reference 60 (Bath TS, Olsen JV. Offline High pH Reversed-Phase Peptide Fractionation for Deep Phosphoproteome Coverage. *Methods Mol Biol* 1355, 179-192 (2016)).

However neither in the text of the manuscript nor in the figure description do they tell us that Fig. S1a is copied directly from Fig. 2 of Bath and Olsen (2016) (attached to this review). This would appear to violate Nature Communications policy as it does not provide appropriate and unambiguous attribution. Moreover, in this case since it is a verbatim copy of part of an image from another paper, permission to reproduce that image would typically need to be obtained from the copyright holder.

We thank this reviewer for pointing this out and apologize for this oversight. We now generated a new Supplementary Figure 1a as the flowchart of experimental design for proteomic studies, which does not contain any published image (as shown in the figure on the right).